# QKFormer: Hierarchical Spiking Transformer using Q-K Attention

**Chenlin Zhou**[1]*, **Han Zhang**[1,2]*, **Zhaokun Zhou**[1,3]*, **Liutao Yu**[1], **Liwei Huang**[1,3],
**Xiaopeng Fan**[1,2], **Li Yuan**[1,3], **Zhengyu Ma**[1]†, **Huihui Zhou**[1]†, **Yonghong Tian**[1,3]

[1]Pengcheng Laboratory     [2]Harbin Institute of Technology     [3]Peking University

## Abstract

Spiking Transformers, which integrate Spiking Neural Networks (SNNs) with
Transformer architectures, have attracted significant attention due to their potential
for low energy consumption and high performance. However, there remains a
substantial gap in performance between SNNs and Artificial Neural Networks
(ANNs). To narrow this gap, we have developed QKFormer, a direct training spik-
ing transformer with the following features: i) *Linear complexity and high energy
efficiency*, the novel spike-form Q-K attention module efficiently models the token
or channel attention through binary vectors and enables the construction of larger
models. ii) *Multi-scale spiking representation*, achieved by a hierarchical structure
with the different number of tokens across blocks. iii) *Spiking Patch Embedding
with Deformed Shortcut (SPEDS)*, enhances spiking information transmission
and integration, thus improving overall performance. It is shown that QKFormer
achieves significantly superior performance over existing state-of-the-art SNN
models on various mainstream datasets. Notably, with comparable size to Spik-
former (66.34 M, 74.81%), QKFormer (64.96 M) achieves a groundbreaking top-1
accuracy of **85.65%** on ImageNet-1k, substantially outperforming Spikformer by
**10.84%**. To our best knowledge, this is the first time that directly training SNNs
have exceeded 85% accuracy on ImageNet-1K. The code and models are available
at https://github.com/zhouchenlin2096/QKFormer.

## 1 Introduction

Regarded as the third generation of neural networks [1], the brain-inspired Spiking Neural Networks
(SNNs) are potential competitors to Artificial Neural Networks (ANNs) due to their high biological
plausibility and high energy efficiency attributed to their event-driven properties [2]. Transformer,
originally designed for natural language processing [3], has flourished in various computer vision
tasks, including image classification [4, 5], object detection [6, 7, 8] and semantic segmentation
[9, 10]. Spiking Transformers (Transformer-based SNNs) [11, 12, 13, 14, 15], which integrate spiking
neural networks with transformer architecture, have attracted significant attention. This innovative
combination provides great potential to develop advanced AI algorithms with high performance and
low energy consumption.

As the architecture of the transformers is essential to the model's performance [4, 5, 16, 8, 10],
designing new architectures for transformer-based SNNs is quite challenging in terms of space
requirements for the following reasons [11, 13, 15]. i). Spiking Self Attention (SSA) [11], the core
module of spiking transformers, encodes Query, Key, and Value with sparse spikes. However, the

---

*Equal
†Corresponding author

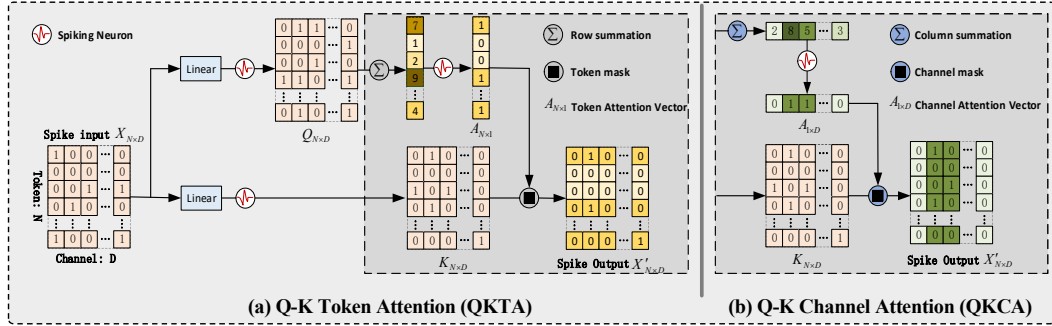

Figure 1: Illustration of Q-K attention with the two versions of Q-K token attention (QKTA) and Q-K channel attention (QKCA). The inputs are binary spikes and there are only sparse additions and mask operations in Q-K attention. As a spike-driven module, Q-K attention efficiently models the token or channel attention through spike-form binary vectors, performing linear complexity to #tokens (or #channels) and high energy efficiency. Spiking Neuron (SN) in this work adopts the Leaky-Integrate-and-Fire (LIF) model, which is shown in Appendix. 7.1.

computational complexity (especially space complexity) of SSA scales quadratically to the number of tokens (#tokens), and is the main obstacle to explore architecture that incorporate multi-level features. ii). SNNs process data across the time domain, necessitating a high level of computational and memory resources. This combination leads to considerable consumption of computational resources, making the training process highly demanding in terms of both memory and processing power.

To address these issues, we propose QKFormer with three innovations. i) Q-K attention with linear complexity and high energy efficiency. ii) A hierarchical architecture with decreasing number of tokens across blocks. iii) A novel patch embedding with deformed shortcut module. The linear complexity of Q-K attention is originated from the binary spike-form vector attention. This design lower the energy consumption and the space requirement. The hierarchical architecture starts from small patches and gradually merges neighboring patches in deeper spiking transformer layers with gradually decreasing #tokens, which enables multi-level spiking feature representation and benefits the model performance. The patch embedding with deformed shortcut facilitates spiking information transmission and integration. These merits make QKFormer achieve state-of-the-art performance in the SNN domain, in contrast to the previous transformer-based SNNs with spiking feature maps of a single resolution. Our main contributions are as follows:

1) We develop a novel spike-form Q-K attention mechanism, tailor-made for the spatio-temporal spiking patterns of SNNs, which can easily model the importance of token or channel dimensions with binary values. The Q-K attention has linear complexity to #tokens (or #channels) and only adopts two spike-form components: Query ($\mathbf{Q}$) and Key ($\mathbf{K}$).

2) We design a versatile and powerful Spiking Patch Embedding with Deformed Shortcut (SPEDS) module, which enhances spiking information transmission and integration thus improving the performance of spiking transformers significantly.

3) We build a direct-training hierarchical spiking transformer with different number of tokens across blocks, incorporating Q-K attention and SPEDS, named QKFormer. This marks the effective exploration of hierarchical spiking representation in Transformer-based SNNs.

4) Extensive experiments show that the proposed model outperforms the state-of-the-art (SOTA) SNNs on various static and neuromorphic datasets. Notably, QKFormer has achieved a significant milestone, surpassing **85%** top-1 accuracy on ImageNet with 4 time steps using the direct training approach for the first time.

## 2 Related Work

**Learning Methods of Spiking Neural Networks.** At present, there are mainly two ways to obtain trained SNNs. One involves converting pre-trained ANNs to SNNs (ANN2SNN) [17, 18, 19, 20, 21, 22, 23], replacing the ReLU activation function in ANN with spiking neurons. However, This converted SNN suffers from long converting time steps and constraints on the original ANN design.

Another method is to directly train SNNs[24], using surrogate gradient[25, 26, 27, 28] to address the non-differentiability of spike excitation function during backpropagation. The direct training method has received more attention due to its low latency and supporting flexible architectural exploration.

**Direct Trained SNN Models.** [28] proposed the Spike-Element-Wise block, which further addressed gradient explosion and gradient vanishing problems, and prolonged the directly trained SNNs beyond a depth of 100 layers with 69.26% accuracy on ImageNet-1k. Spikformer [11] designed a novel spike-form self-attention named Spiking Self Attention (SSA), using sparse spike-form Query, Key, and Value without softmax operation, which was used to construct the Spikformer. Spikformer achieved 74.81% accuracy on ImageNet-1k with 4 time steps, showing the great potential of transformer-based SNNs for the first time. Spikingformer [12] modified Spikformer with a pre-activation shortcut, which can avoid the floating-point multiplications in synaptic computing and has a lower firing rate. Spikingformer achieved 75.85% accuracy on ImageNet-1k. [13] designed a novel Spike-Driven Self-Attention (SDSA), which used only masks and addition operations without any multiplication, thus significantly reducing the computation energy compared to the vanilla self-attention. In addition, the proposed Spike-driven Transformer based on SDSA has achieved 77.07% on ImageNet-1k. However, all of these SNN models above remain a large performance gap compared with ANN.

## 3 Method

### 3.1 Preliminary

**Vanilla Self Attention.** Vanilla self-attention (VSA) [3] in transformers has three floating-point key components: query ($\mathbf{Q}_{\mathcal{F}}$), key ($\mathbf{K}_{\mathcal{F}}$), value ($\mathbf{V}_{\mathcal{F}}$) which are calculated by learnable linear matrics and input $\mathbf{X}$. The calculation of VSA can be formulated as follows:

$$\mathbf{Q}_{\mathcal{F}}, \mathbf{K}_{\mathcal{F}}, \mathbf{V}_{\mathcal{F}} = \mathbf{X}(\mathbf{W}_Q, \mathbf{W}_K, \mathbf{W}_V), \tag{1}$$

$$\text{VSA}\left(\mathbf{Q}_{\mathcal{F}}, \mathbf{K}_{\mathcal{F}}, \mathbf{V}_{\mathcal{F}}\right) = \text{Softmax}\left(\frac{\mathbf{Q}_{\mathcal{F}}\mathbf{K}_{\mathcal{F}}^{\mathrm{T}}}{\sqrt{d}}\right)\mathbf{V}_{\mathcal{F}}, \tag{2}$$

where $\mathcal{F}$ denotes the floating-point form. Both floating-point matrix multiplication and softmax operation which contains exponent calculation and division, do not align with the properties of SNNs.

**Spiking Self Attention.** Spikformer [11] demonstrated a novel spike-form self-attention named Spiking Self Attention (SSA), using sparse spike-form $\mathbf{Q}, \mathbf{K}, \mathbf{V}$ without softmax operation and floating-point matrix multiplication. The calculation process of SSA is formulated as follows:

$$\mathbf{I} = \text{SN}_I\left(\text{BN}_I\left(\mathbf{X}(\mathbf{W}_I)\right)\right), \mathbf{I} \in (\mathbf{Q}, \mathbf{K}, \mathbf{V}), \tag{3}$$

$$\text{SSA}'(\mathbf{Q}, \mathbf{K}, \mathbf{V}) = \text{SN}\left(\mathbf{Q}\mathbf{K}^{\mathrm{T}}\mathbf{V} * s\right), \tag{4}$$

where $\mathbf{Q}, \mathbf{K}, \mathbf{V} \in \mathcal{R}^{T \times N \times D}$, the spike-form $\mathbf{Q}, \mathbf{K}, \mathbf{V}$ are computed by learnable linear layers. $s$ is a scaling factor. SN means spiking neuron layer. The calculation of SSA avoids floating-point multiplication, meeting the property of SNNs.

### 3.2 Q-K Attention

An overview of Q-K attention is shown in Figure 1. Both VSA and SSA use three key components and have $O(N^2d)$ or $O(Nd^2)$ computational complexity, while our proposed Q-K Attention which has linear complexity and only uses two spike-form components: $\mathbf{Q}$ and $\mathbf{K}$, which are produced through learnable linear matrics.

$$\mathbf{Q} = \text{SN}_Q\left(\text{BN}\left(\mathbf{X}\mathbf{W}_Q\right)\right), \mathbf{K} = \text{SN}_K\left(\text{BN}\left(\mathbf{X}\mathbf{W}_K\right)\right), \tag{5}$$

where $\mathbf{X}$ is the input spiking map. According to the detailed calculation mechanism of $\mathbf{Q}, \mathbf{K}$, Q-K Attention can be divided into Q-K Token Attention (QKTA) and Q-K Channel Attention (QKCA).

**Q-K Token Attention.** We here assume $T = 1$ and single head attention for mathematical description. After obtaining spike-form $\mathbf{Q}, \mathbf{K} \in \mathcal{R}^{T \times N \times D}$, both $\mathbf{Q}$ and $\mathbf{K}$ can be formed as a spike matrix $N \times D$ ($N$ is the token number, $D$ is the channel number). QKTA process can be formulated as follows:

$$\mathbf{A}_t = \text{SN}(\sum_{i=0}^{D} \mathbf{Q}_{i,j}), \quad \mathbf{X}' = \mathbf{A}_t \otimes \mathbf{K}, \tag{6}$$

Table 1: Computational complexity comparison. $N$ is the token number, $D$ is the channel number.

| Methods | VSA [3] | SSA [11] | SDSA [13] | QKTA | QKCA |
|---|---|---|---|---|---|
| Time complexity | $O(N^2 D)$ | $O(N^2 D)$ | $O(ND)$ | $O(D)$ | $O(N)$ |
| Space complexity | $O(N^2 + ND)$ | $O(N^2 + ND)$ | $O(ND)$ | $O(N)$ | $O(D)$ |

where $\mathbf{A}_t$ is the $N * 1$ token attention vector, which models the binary importance of different tokens. $\mathbf{A}_t$ is a spike-form vector, which is obtained by addition operations (row summation) of $\mathbf{Q}$ spike matrix and a following spiking neuron. $\otimes$ is the Hadamard product between spike tensors, which is equivalent to the mask operation. We apply the spike-form token attention vector $\mathbf{A}_t$ to the $\mathbf{K}$ spike matrix through the column mask operation (token mask), to obtain the output $\mathbf{X}'$ of QKTA.

**Q-K Channel Attention.** The calculation process of Q-K channel attention is similar to the previous Q-K token attention, and can be formulated as : $\mathbf{A}_c = \mathrm{SN}(\sum_{j=0}^{N} \mathbf{Q}_{i,j}), \quad \mathbf{X}' = \mathbf{A}_c \otimes \mathbf{K}$, where $\mathbf{A}_c$ is the $1 * D$ channel attention vector, which models the binary importance of different channels. $\mathbf{A}_t$ is a spike-form vector, which is obtained by addition operations (column summation) of Q spike matrix and a following spiking neuron. Then, the output $\mathbf{X}'$ of Q-K Channel Attention is obtained by the row mask operation (channel mask) between $\mathbf{A}_t$ and $\mathbf{K}$.

$$\mathbf{X}'' = \mathrm{SN}\left(\mathrm{BN}\left(\mathrm{Linear}\left(\mathbf{X}'\right)\right)\right). \tag{7}$$

As shown in Formula.7, a post-linear layer is also required after obtaining $\mathbf{X}'$ of Q-K Token or Channel Attention. In addition, the channel dimension is $D/h$ in the multi-head Q-K attention, where $h$ is the head number. In this work, the spiking neuron uses the LIF model [28]. Same with [11], time step $T$ is an independent dimension for the spiking neuron layer. In other layers, it is merged with the batch size. We exploit QKTA in our experiments by default.

**Linear Computational Complexity of Q-K Attention.** As shown in Table 1, the time complexity of Q-K attention varies based on the implementation approach. Specifically, when utilizing spike-form broadcasted element-wise multiplication, $\otimes$, the time complexity can reach up to $O(N * D)$. When applying mask operation, the time complexity of Q-K attention is only $O(N)$ or $O(D)$. The space complexity of Q-K attention with the whole process is $O(N * D)$ at most, which is caused by the self-storage consumption Q and K matrix. In terms of the space complexity of attention operation, Q-K attention only requires an extra $1 * D$ or $N * 1$ space to store the attention vector with the space complexity of $O(N)$ or $O(D)$. The linear complexity of Q-K attention makes it possible to successfully explore the large-scale hierarchical architecture SNN model.

**High Energy Efficiency of Q-K Attention.** As a spike-driven attention module, the linear multiplication is transformed into sparse addition. Mask operation can be implemented on neuromorphic chips through addressing algorithms [29] or AND logic operations[30] with negligible power consumption. Compared with SSA, Q-K attention is much more energy-efficient, which comes from the following reasons: i) Q-K attention only adopts two spike-form components for spike [0, 1] operation without the V input and thus has less synaptic computing. ii) Q-K attention has much fewer spiking matrix operations due to its linear complexity of $O(N)$ or $O(D)$. iii) Q-K attention discards the scale operation of SSA, which leads to reduced power consumption further.

## 3.3 No Scaling Factors in Q-K Attention

In VSA [3], assume that $\mathbf{q}_i$ $\left(\mathbf{q}_i \in R^{1 \times d}, \mathbf{Q} \in R^{m \times d}\right)$ and $\mathbf{k}_i$ $\left(\mathbf{k}_i \in R^{1 \times d}, \mathbf{K} \in R^{m \times d}\right)$ are independent random variables with a mean of 0 and a variance of 1, then each element in the product of $\mathbf{Q}\mathbf{K}^{\mathrm{T}}$ has mean 0 and variance $d$. The variance magnitude of $\mathbf{Q}\mathbf{K}^{\mathrm{T}}$ grows with the embedding dimension $d$, which can result in gradient vanishing issues after softmax operation. Therefore, The product of matrices $\mathbf{Q}$ and $\mathbf{K}$ in VSA [3] is scaled by a factor $\frac{1}{\sqrt{d}}$ in Eq. 2 to normalize the product to variance 1. Though the softmax function is not adopted due to its non-spike operations (division, exponential operation) in SNNs, SSA-based [11] SNNs will suffer obvious performance degradation even cannot converge without scaling because the variance of $\mathbf{Q}\mathbf{K}^{\mathrm{T}}\mathbf{V}$ output is too large (Assuming that all the spiking elements are independent random variables and subject to Bernoulli Distribution). However, Q-K attention can discard scaling operations thus reducing power consumption because the variance of Q-K attention is much smaller than SSA (e.g. the max theoretical variance of Q-K token attention is only about 1 / 200 of SSA). The detailed analysis can be found in the Appendix.7.2 and Section.4.3.

### 3.4 QKFormer

As the computational complexity (especially space complexity) of SSA is quadratic to #tokens, previous direct training spiking transformers are all limited to straight-through structures. Combining SSA with hierarchical architecture directly will lead to memory explosion easily when training spiking transformers. To overcome these issues, we proposed a hierarchical spiking transformer based on Q-K attention, named QKFormer, which constructs hierarchical spiking feature maps with linear computational complexity to #tokens or #channels.

**Overall Hierarchical Architecture.** The overview of QKFormer is presented in Figure 2. The input form can be formulated as $(T_0 \times H \times W \times n)$. In static RGB image datasets, $T_0 = 1$ and $n = 3$. In temporal neuromorphic datasets, the input $T_0 = T$, while $n = 2$. In our implementation, we use a patch size of $4 \times 4$ and thus the input feature dimension $(4 \times 4 \times n)$ of each patch is projected into a spike-form arbitrary dimension (denoted as $C$) in Spiking Patch

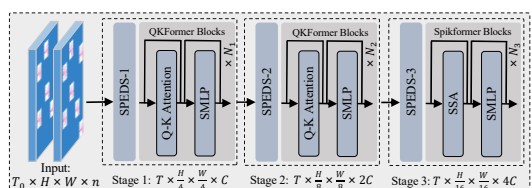

Figure 2: The overview of QKFormer, a hierarchical spiking transformer with Q-K attention.

Embedding with Deformed Shortcut 1 (SPEDS-1), which together with the following QKFormer blocks are referred to as "Stage 1". The number of tokens in Satge 1 is $(\frac{H}{4} \times \frac{W}{4})$. To produce a hierarchical spiking representation, the number of tokens is reduced in SPEDS-2 and SPEDS-3 as the network goes deeper. Both SPEDS-2 and SPEDS-3 reduce the number of tokens by a patch size of $2 \times 2$ ($2 \times$ downsampling of resolution), and the number of channels is transformed into $2C$ and $4C$, respectively. We denote the SPEDS-2 and the following QKFormer blocks as "Stage 2", which reduces the number of tokens $(\frac{H}{8} \times \frac{W}{8})$. While SPEDS-3 and the following Spikformer or QKormer blocks are referred to as "Stage 3" with $(\frac{H}{16} \times \frac{W}{16})$ tokens. The number of spiking transformer blocks (QKFormer or Spikformer) in each stage are $N_1$, $N_2$, and $N_3$, respectively. These stages jointly produce a hierarchical spike-form representation.

**Mixed Spiking Attention Integration.** In the former stage of a hierarchical architecture model, the number of channels is small while the number of tokens is large. In the latter stage, the channel number is large while the token number is small. Thus it leads to suboptimal performance when we only use a single type of Q-K attention in a hierarchical architecture model. Therefore, we use mixed spiking attention integration in QKFormer. QKTA is conducted in the former stage in hierarchical architecture, and we could choose QKCA or SSA in the latter stage. In the subsequent experiments, we use SSA in the last stage of QKFormer and QKTA in the former stages by default.

**QKFormer Blocks.** Similar to the standard transformer encoder block, a QKFormer block contains a Q-K Attention module (QKTA or QKCA) and a Spiking MLP (SMLP) block, which can be formulated as follows:

$$\mathbf{X}'_l = \mathrm{QKTA}\left(\mathbf{X}_{l-1}\right) + \mathbf{X}_{l-1}, \mathbf{X}'_l \in R^{T \times N \times D}, \tag{8}$$

$$\mathbf{X}_l = \mathrm{SMLP}\left(\mathbf{X}'_l\right) + \mathbf{X}'_l, \mathbf{X}_l \in R^{T \times N \times D}. \tag{9}$$

At last, a fully connected layer is used as the classifier behind the last block.

### 3.5 Spiking Patch Embedding with Deformed Shortcut.

Residual shortcuts in SNNs [28] can implement identity mapping, which reduces information loss (facilitates information transmission and integration) in spike communication, thus ensuring the network can be well-behaved in a depth-insensitive way. Previous spiking transformers [11, 12, 13] use the residual shortcuts to achieve identity mapping, mainly focusing on the spiking attention block and spiking MLP block, and lacking identity mapping in patch embedding across the downsampling block. The input and output of a spiking patch embedding block in QKFormer have different channel and token numbers. To realize residual learning in spiking patch embedding, we can perform a lightweight linear projection $\mathbf{W}_d$ in the shortcut connections to match the channel and token numbers, thus realizing the identity mapping cross downsampling blocks in spiking patch embedding. Given the input spiking map $\mathbf{X}$, the process of patch embedding can be formulated as follows:

$$\mathbf{Y} = \mathcal{F}\left(\mathbf{X}, \{\mathbf{W}_i\}\right) + \mathrm{SN}(\mathbf{W}_d\mathbf{X}). \tag{10}$$

Table 2: Results on ImageNet-1K. Power is calculated as the average theoretical energy consumption when predicting an image from ImageNet test set. The power data for QKFormer and ANNs is evaluated according to Appendix.7.6, and the power data for other works were obtained from related papers. "A2S" denotes "ANN-to-SNN", "HST-$L$-$D$" denotes "Hierarchical Spiking Transformer" with $L$ encoder blocks and $D$ channels. HST-10-768* and HST-10-768** means HST-10-768 with $288^2$ and $384^2$ input size for inference. The top-5 accuracy of QKFormer (HST-10-768**) is 97.74%.

| Methods | Type | Architecture | Input Size | Param (M) | Power (mJ) | Time Step | Top-1 Acc (%) |
|---|---|---|---|---|---|---|---|
| RMP[21] | A2S | VGG-16 | $224^2$ | 39.90 | - | 2048 | 73.09 |
| QCFS[22] | A2S | ResNet-18 | $224^2$ | 11.70 | - | 1024 | 74.32 |
| MST[23] | A2S | Swin Transformer-T | $224^2$ | 28.50 | - | 512 | 78.51 |
| SEW ResNet[28] | SNN | SEW-ResNet-34 | $224^2$ | 21.79 | 4.89 | 4 | 67.04 |
| | SNN | SEW-ResNet-101 | $224^2$ | 44.55 | 8.91 | 4 | 68.76 |
| | SNN | SEW-ResNet-152 | $224^2$ | 60.19 | 12.89 | 4 | 69.26 |
| Spikformer[11] | SNN | Spikformer-8-384 | $224^2$ | 16.81 | 7.73 | 4 | 70.24 |
| | SNN | Spikformer-8-512 | $224^2$ | 29.68 | 11.58 | 4 | 73.38 |
| | SNN | Spikformer-8-768 | $224^2$ | 66.34 | 21.48 | 4 | 74.81 |
| Spikingformer[12] | SNN | Spikingformer-8-384 | $224^2$ | 16.81 | 4.69 | 4 | 72.45 |
| | SNN | Spikingformer-8-512 | $224^2$ | 29.68 | 7.46 | 4 | 74.79 |
| | SNN | Spikingformer-8-768 | $224^2$ | 66.34 | 13.68 | 4 | 75.85 |
| S-Transformer[13] | SNN | S-Transformer-8-384 | $224^2$ | 16.81 | 3.90 | 4 | 72.28 |
| | SNN | S-Transformer-8-512 | $224^2$ | 29.68 | 1.13 | 1 | 71.68 |
| | SNN | S-Transformer-8-512 | $224^2$ | 29.68 | 4.50 | 4 | 74.57 |
| | SNN | S-Transformer-8-768* | $288^2$ | 66.34 | 6.09 | 4 | 77.07 |
| ViT[4] | ANN | ViT-B/16 | $384^2$ | 86.59 | 254.84 | 1 | 77.90 |
| DeiT[32] | ANN | DeiT-B | $224^2$ | 86.59 | 80.50 | 1 | 81.80 |
| | ANN | DeiT-B | $384^2$ | 86.59 | 254.84 | 1 | 83.10 |
| Swin[8] | ANN | Swin Transformer-B | $224^2$ | 87.77 | 70.84 | 1 | 83.50 |
| | ANN | Swin Transformer-B | $384^2$ | 87.77 | 216.20 | 1 | 84.50 |
| **QKFormer** | SNN | HST-10-384 | $224^2$ | 16.47 | 15.13 | 4 | 78.80 |
| | SNN | HST-10-512 | $224^2$ | 29.08 | 21.99 | 4 | 82.04 |
| | SNN | HST-10-768 | $224^2$ | 64.96 | 8.52 | 1 | 81.69 |
| | SNN | HST-10-768 | $224^2$ | 64.96 | 38.91 | 4 | 84.22 |
| | SNN | HST-10-768* | $288^2$ | 64.96 | 64.27 | 4 | 85.25 |
| | SNN | HST-10-768** | $384^2$ | 64.96 | 113.64 | 4 | **85.65** |

In this work, the deformed linear projection $\mathbf{W}_d$ is set as a lightweight convolutional layer with $1 \times 1$ kernel and stride $> 1$, to meet the channel and token numbers of the patch embedding block. The function $\mathcal{F}$ involved in this work is set as {Conv2D-BN-MaxPooling-SN-Conv2D-BN-SN} or {Conv2D-BN-SN-Conv2D-BN-MaxPooling-SN}, while more layers or more variants are possible.

There are mainly two types of residual shortcuts in deep SNNs. Formula.10 shows the patch embedding in the way of activation-before-addition [28, 11]. The other way of the patch embedding with the pre-activation residual shortcut [31, 12, 13] can be formulated as follows:

$$\mathbf{Y} = \text{SN}(\mathcal{G}\,(\mathbf{X}, \{\mathbf{W}_j\}) + \mathbf{W}_d\mathbf{X}), \tag{11}$$

where the function $\mathcal{G}$ correspondingly could be set as {Conv2D-BN-MaxPooling-SN-Conv2D-BN} or {Conv2D-BN-SN-Conv2D-BN-MaxPooling}. The intuitive representation of SPEDS is shown in Appendix 7.4.

In this work, the spiking patch embedding of stage 2 or stage 3 in QKFormer can be formulated as Formula.10. The spiking patch embedding in stage 1 uses an extra {Conv2D-BN-SN} for spiking encoding in front of the block (Formula.10) to transform the non-spike input data into spikes.

Table 3: Comparision on CIFAR10, CIFAR100, DVS128, CIFAR10-DVS. "Param" denotes "Parameter (M)", "Acc" denotes "Top-1 Accuracy (%)", "$T$" denotes "Time Step".

| Method | CIFAR10 | | | CIFAR100 | | | DVS128 | | | CIFAR10-DVS | | |
|---|---|---|---|---|---|---|---|---|---|---|---|---|
| | Param | $T$ | Acc | Param | $T$ | Acc | Param | $T$ | Acc | Param | $T$ | Acc |
| Spikformer [11] | 9.32 | 4 | 95.51 | 9.32 | 4 | 78.21 | 2.57 | 16 | 98.3 | 2.57 | 16 | 80.9 |
| Spikingformer [12] | 9.32 | 4 | 95.81 | 9.32 | 4 | 78.21 | 2.57 | 16 | 98.3 | 2.57 | 16 | 81.3 |
| CML [14] | 9.32 | 4 | 96.04 | 9.32 | 4 | 80.02 | 2.57 | 16 | 98.6 | 2.57 | 16 | 80.9 |
| S-Transformer[13] | 10.28 | 4 | 95.60 | 10.28 | 4 | 78.4 | 2.57 | 16 | **99.3** | 2.57 | 16 | 80.0 |
| STSA[15] | – | – | – | – | – | – | 1.99 | 16 | 98.7 | 1.99 | 16 | 79.93 |
| ResNet-19 (ANN) | 12.63 | 1 | 94.97 | 12.63 | 1 | 75.35 | – | – | – | – | – | – |
| Trasnformer (ANN) | 9.32 | 1 | 96.73 | 9.32 | 1 | 81.02 | – | – | – | – | – | – |
| **QKFormer** | 6.74 | 4 | **96.18** | 6.74 | 4 | **81.15** | 1.50 | 16 | 98.6 | 1.50 | 16 | **84.0** |

## 4 Experiments

### 4.1 Results on ImageNet-1k Classification

**Experimental Setup on ImageNet.** In this experiment, we use AdamW as the optimizer, which is adopted with a base learning rate of $6 \times 10^{-4}$. The actual learning rate was calculated as BatchSize/256 multiplied by the base learning rate. The batch size is set to $512$, which is realized by accumulated gradient iterations [33] and distributed across 8 Nvidia V100 GPUs. We trained QKFormer for 200 epochs. In addition, following DeiT [32], data augmentation techniques including RandAugment [34], random erasing [35], and stochastic depth [36] are employed in this study. The number of blocks in the three stages is set as {1, 2, 7} respectively.

**Main Results on ImageNet.** The experimental results demonstrate the superior performance of our proposed QKFormer, surpassing previous works' performance by a large margin (Table 2). QKFormer (**64.96 M**) achieves **85.65%** top-1 accuracy and **97.74%** top-5 accuracy on ImageNet. To begin with, we compare our model with the baseline spiking transformer (i.e., Spikformer [11]). Our QKFormer models have slightly fewer parameters but much higher performance. For example, our QKFormer (64.96 M, 85.65%) significantly outperforms Spikformer (66.34 M, 74.81%) by **10.84%**. In addition, compared with SDSA, our Q-K attention has lower computational complexity (shown in Table 1) and our QKFormer has much higher performance than S-Transformer (built by SDSA) [13]. In detail, QKFormer outperforms S-Transformer by 7.55%, 7.47%, and 8.58% respectively on three models with comparable #parameters. Finally, Our QKFormer outperforms the SOTA ANN-to-SNN model MST [23] by 7.14% and has much fewer time steps meanwhile. To our best knowledge, this is the first time that a direct training SNN model has achieved an accuracy of over **85%** on ImageNet-1k.

**Comparing with ANN Models on ImageNet.** Our QKFormer is an event-driven SNN model, whose output is in binary form (either 0 or 1), the multiplications of activations and weights can be transformed into sparse addition, thus enjoying high energy efficiency. It should be noted that hierarchical architecture will lead to the power increment of QKFormer. This is still very cost-effective compared with ANN models. For instance, QKFormer (**64.96M, 85.65%, SNN, 113.64mJ**) Vs. Swin Transformer (**88M, 84.5%, ANN, 216.20mJ**) [8] Vs. DeiT-B (86M, 83.1%, ANN, 254.84mJ) [32] Vs. ViT (85.59M, 77.9%, ANN, 254.84mJ). [4]. Under the same experiment conditions without pre-training or extra training data, our QKFormer has surpassed the most well-known Transformer-based ANNs in performance while maintaining high energy efficiency.

### 4.2 Results on CIFAR and Neuromorphic Datasets

**CIFAR Classification.** In this experiment, the QKFormer is trained for 400 epochs with a batch size of 64 following previous works: Spikformer [11], Spikingformer [12]. Following Spikformer, we use 4 blocks in QKFormer in total, which are distributed {1, 1, 2} in three stages. Due to the hierarchical architecture design, our QKFormer model has only 6.74 M parameters in this case. The results on CIFAR datasets are shown in Table 3. For CIFAR10, our model achieved **96.18%** accuracy with **6.74 M** parameters. Our proposed QKFormer outperforms Spikformer by 0.67% and reduces 2.58 M

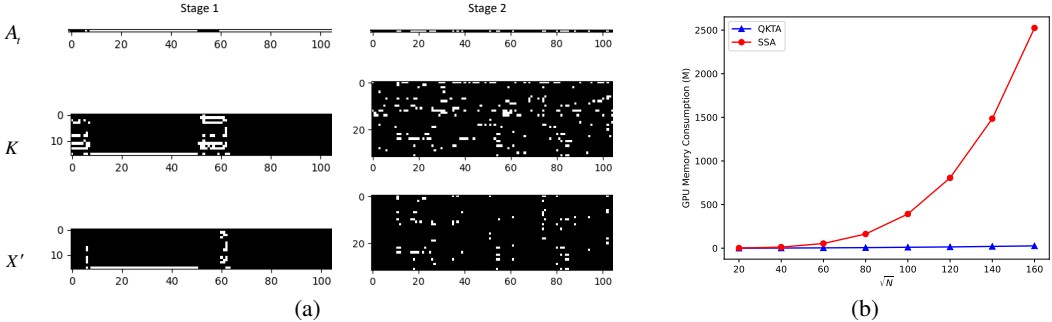

(a)                                                            (b)

Figure 3: The visualization and memory consumption of QKTA. (a) is the visualization of Q-K token attention. The white dot means value 1, while the black one means value 0. (b) shows the comparison of memory costs between QKTA and SSA under different token numbers. $N$ is the token number.

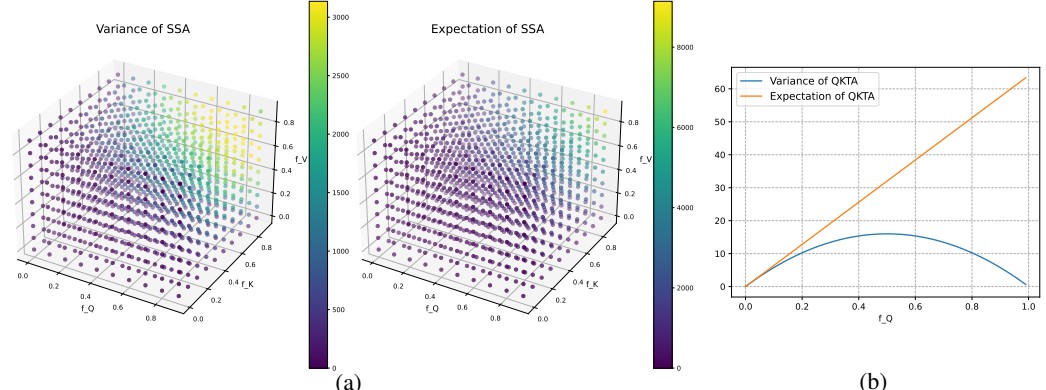

(a)                                                            (b)

Figure 4: (a) shows the variance and expectation of SSA, (b) shows the variance and expectation of QKTA. Assume that all the spike elements (either 0 or 1) in SSA and QKTA are independent random variables and subject to Bernoulli distribution.

parameters meanwhile. For CIFAR100, our model achieved **81.15%** with 6.74 M parameters. Our proposed QKFormer outperforms Spikformer by **2.94%** and reduces 2.58 M parameters meanwhile.

**Neuromorphic Classification.** We compare our method with SOTA methods on both CIFAR10-DVS and DVS-Gesture datasets. In this experiment, We utilize a mini QKFormer model with 1.50 M parameter, which has {0, 1, 1} blocks in three stages. The max patch embedding dimension is set to 256. The training process involves 200 epochs for DVS128 Gesture and 106 epochs for CIFAR10-DVS. The number of time steps of the spiking neuron is 10 or 16. The experimental results of temporal neuromorphic classification are presented in Table 3. For DVS128-Gesture dataset, our model with 1.50 M parameters achieves 98.6% accuracy using 16 time steps and 98.3% accuracy using 10 time steps. For CIFAR10-DVS dataset, our model achieves **84.0%** accuracy with only **1.50 M** parameters using 16 time steps. Our proposed QKFormer significantly outperforms Spikformer by **3.1%** while reducing 1.07 M parameters. In addition, our model with 10 time steps achieves 83.8% accuracy, which outperforms Spikformer by **4.9%** and outperforms the SOTA model (Spikingformer) by 3.9%.

## 4.3 Analyses on Q-K Attention

**Attention Visualization.** In this part, we visualize the Q-K token attention (Stage 1 and Stage 2 of the QKFormer model) on ImageNet. As shown in Figure 3(a), $\mathbf{A}_t$ is the $N * 1$ token attention vector, and $\mathbf{X}'$ is the output of the attention process, which is obtained by the mask operation between matrix $\mathbf{K}$ and attention vector $\mathbf{A}_t$. Specifically, the longitudinal axis denotes the channel index of one head, while the horizontal axis denotes the token index. The #tokens in stage 1 and stage 2 are $56^2$ and $28^2$, respectively. To facilitate visualization, we choose a continuous segment with a length of 100 extracted from the whole token vector. The visualization shows Q-K attention can lead to high sparsity of spikes.

**Memory Consumption.** In this experiment, we compare the memory consumption between QKTA (Formula.6) and SSA (Formula.4) under different token numbers. We calculate the memory consumption of a QKTA and an SSA on a GPU by forwarding the input tensor $(T, B, C, N)$. To facilitate the statistics of the impact of #tokens $N$ on memory consumption, the #channels $C$ is set to 256, and the time step $T$ and batch size $B$ are set to 1. The experiment result is shown in Figure 3(b). With the increment of #Tokens, SSA consumes much more GPU memory than QKTA, of which the complexity is linear to #Tokens. For example, SSA consumes about $10\times$ GPU memory than QKTA when $\sqrt{N} = 50$.

**Spike Firing Rates in QKFormer Blocks.** In this experiment, we calculate the spike firing rates of QKFormer blocks of the trained QK-Former (64.9M) on the ImageNet-1K test set with the $224 \times 224$ input resolution. The average spike firing rates of the QKFormer blocks in Stage1 and Stage2 are shown in Table 4. Note that the spike-form $\mathbf{X}'$ is obtained by column mask operation (token mask) between $\mathbf{A}_t$ and $\mathbf{K}$. In fact, the summation operation in the Q-K attention causes $\mathbf{Q}$ to become significantly

Table 4: Spike firing rates in QKFormer blocks.

| QKFormer Block | | Stage1 (fr) | Stage2 (fr) |
|---|---|---|---|
| QKTA | $\mathbf{Q}$ | 0.0432 | 0.0231 |
| | $\mathbf{K}$ | 0.1784 | 0.0847 |
| | $\mathbf{A}_t$ | 0.3477 | 0.2655 |
| | $\mathbf{X}'$ | 0.0832 | 0.0350 |
| | $\mathbf{X}''$ | 0.1478 | 0.0577 |
| SMLP | Layer1 | 0.0518 | 0.0246 |
| | Layer2 | 0.2733 | 0.1869 |

sparser compared to $K$ when the network converges. Specifically, $\mathbf{Q}$ in stage 1 has a fire rate of 0.0432, while $\mathbf{K}$ has 0.1784. After the accumulation operation along $D/h$ of the multi-head QKTA version, the LIF neuron ($\mathbf{A}_t$) exhibits a typical averaged fire rate of 0.3477.

**The Variance and Expectation of QKTA.** Figure.4 visualize the variance and expectation of QKTA (Formula.15 and 16 in Appendix.7.2) and SSA (Formula.19 and 20 in Appendix.7.2). $N$ is set as 196 and $d$ is set as 64, respectively. We can find SSA has a much larger variance and expectation than QKTA on the whole. For example, the maximum theoretical variance of QKTA is 16, but the maximum theoretical variance of SSA is over 3000. This is the main reason that Q-K attention can discard scaling operations thus reducing power consumption, but SSA can not.

## 4.4 Ablation Study

**SPEDS Module.** In this experiment, We replaced the Spiking Patch Splitting (SPS) module in Spikformer with Spiking Patch Embedding with Deformed Shortcut (SPEDS) module, while other conditions remain unchanged. The results (Table 5) show that the SPEDS module is essential to QKFormer on both static and neuromorphic datasets. In addition, the addition of SPEDS to Spikformer leads to great gains in CIFAR100 (+2.05%) and CIFAR10-DVS (+1.30%), which further verified the effectiveness of SPEDS.

Table 5: Ablation studies of SPEDS module.

| Model | CIFAR100 (Acc) | CIFAR10-DVS (Acc) |
|---|---|---|
| QKFormer (QKTA + SSA, baseline) | 81.15% | 84.00% |
| QKFormer (QKTA + SSA, w/o SPEDS) | 80.08% | 83.40% |
| Spikformer (SSA, w/o scaling) | 76.95% | 79.30% |
| Spikformer (SSA) | 78.21% | 80.90% |
| Spikformer (SSA) + SPEDS | 80.26% | 82.20% |

Table 6: Ablation studies of Q-K Attention.

| Model | CIFAR100 (Acc, Param) | CIFAR10-DVS (Acc, Param) |
|---|---|---|
| QKFormer (QKTA + SSA, baseline) | 81.15%, 6.74M | 84.00%, 1.50M |
| QKFormer (QKCA + SSA) | 81.07%, 6.74M | 84.30%, 1.50M |
| QKFormer (QKTA + QKCA) | 81.04%, 6.44M | 83.10%, 1.44M |
| QKFormer (SSA) | 81.23%, 6.79M | 84.10%, 1.52M |
| QKFormer (QKCA) | 81.00%, 6.44M | 80.70%, 1.44M |
| QKFormer (QKTA) | 79.09%, 6.44M | 80.70%, 1.44M |

**Mixed Spiking Attention Integration with Q-K Attention.** In this part, we show different integration strategies of QKCA, QKTA, and SSA. The baseline is our QKFormer (QKTA + SSA, 6.70M). The experimental results (Table 6) show that using a single type of Q-K attention (QKTA or QKCA only) in a hierarchical architecture model leads to suboptimal performance. In particular, the performance decline in QKTA is more obvious. While the mixed spiking attention solutions, such as QKFormer(QKTA + QKCA), QKFormer(QKTA + SSA), and QKFormer(QKCA + SSA) can achieve comparable performance to QKFormer(SSA) while requiring fewer parameters and much fewer memory resources (Figure 3(b)). Consequently, the mixed spiking attention solutions are well-suited for larger architectures and more challenging scenarios when considering both computational efficiency and performance.

**Residual Connection (RC) & Spiking Neuron (SN) & Time Step (TS).** The experimental results are shown in Table 7. In this block, we replaced the Activation-Before-Addition (ABA) [28, 11] residual connection of QKFormer with the Pre-activation (PA) [31, 37] way, and the performance slightly improved. In addition, we replaced the LIF spiking neuron with Integrate-and-Fire (IF) and Parametric-Leaky-Integrate-and-Fire (PLIF) [38], which led to slight performance degradation. The accuracy regarding different simulation time steps of QKFormer is shown in the last column. When we increase the simulation time steps, the performance of QKFormer can be further improved. Specifically, QKFormer achieves 81.30 % accuracy on CIFAR 100 when T=6.

Table 7: Ablation studies of RC, SN, TS.

| Model | CIFAR100 (Acc) |
| --- | --- |
| QKFormer (baseline) | 81.15% |
| QKFormer (ABA->PA) | 81.18% |
| QKFormer (LIF->IF) | 80.95% |
| QKFormer (LIF->PLIF) | 81.12% |
| QKFormer (T=1) | 78.51% |
| QKFormer (T=2) | 80.08% |
| QKFormer (T=4) | 81.15% |
| QKFormer (T=6) | 81.30% |

## 5   Conclusion

In this work, we design a novel spike-form Q-K attention considering the properties of SNNs, which can easily model the importance of token or channel dimensions through binary vectors. Q-K attention has linear complexity to #tokens (or #channels) and only adopts two spike-form components: Query (**Q**) and Key (**K**). We design a versatile and powerful Spiking Patch Embedding with Deformed Shortcut (SPEDS), which enhances spiking information transmission and integration, thus improving the performance of spiking transformers. In addition, we develop a hierarchical spiking transformer based on the proposed Q-K attention and SPEDS in a direct training way, named QKFormer, which marks the effective exploration of hierarchical spiking representation in Transformer-based SNNs. Extensive experiments show that the proposed model achieves SOTA performance on both static and neuromorphic datasets. Note that QKFormer achieved top-1 accuracy beyond 85% on ImageNet-1k with 4 time steps using the direct training way for the first time. With its powerful performance, we aim for our investigations to instill optimism in the application of SNNs.

**Limitation.** Currently, our model is limited to image / DVS classification tasks. We will extend this work to more tasks, such as segmentation, detection, and language tasks, to test the generalizability in the further. In addition, we will explore efficient and high-performance network architectures with fewer time steps based on Q-K attention and other efficient modules, to further reduce the training consumption.

## 6   Acknowledgments and Disclosure of Funding

This work is supported by grants from the National Natural Science Foundation of China (62236009, 62206141, 62027804, and 62425101), and the major key project of the Pengcheng Laboratory (PCL2021A13). Computing support was provided by Pengcheng Cloudbrain.

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

# 7 Appendix

## 7.1 Spiking Neuron Model

Spiking neuron is the fundamental unit of SNNs, we choose the Leaky Integrate-and-Fire (LIF) model as the spiking neuron in our work. The dynamics of a LIF neuron can be formulated as follows:

$$H[t] = V[t-1] + \frac{1}{\tau}\left(X[t] - (V[t-1] - V_{\text{reset}})\right), \tag{12}$$

$$S[t] = \Theta\left(H[t] - V_{th}\right), \tag{13}$$

$$V[t] = H[t](1 - S[t]) + V_{\text{reset}}\, S[t], \tag{14}$$

where $\tau$ is the membrane time constant, and $X[t]$ is the input current at time step $t$. When the membrane potential $H[t]$ exceeds the firing threshold $V_{th}$, the spiking neuron will trigger a spike $S[t]$. $\Theta(v)$ is the Heaviside step function, which equals to 1 when $v \geq 0$ and 0 otherwise. $V[t]$ represents the membrane potential after the triggered event, which equals to $H[t]$ if no spike is generated and otherwise equals to the reset potential $V_{reset}$.

## 7.2 Q-K Attention Vs. SSA in Scaling Factors

**Mathematical Characteristics of Q-K Attention.** All the elements in Q-K attention are spike-form, thus we assume that each $Q_{i,j}[t]$ are independent random variables and subject to Bernoulli distribution $B(f_Q)$. $f_Q$ is the average firing rate of **Q**. The expectation and variance of Q-K attention can be formulated as:

$$\text{E}\left(\text{QKTA}\right) = \text{E}\left(\sum_{i=0}^{d} Q_{i,j}[t]\right) = \sum_{i=0}^{d} \text{E}\left(Q_{i,j}[t]\right) = df_Q, \tag{15}$$

$$\text{Var}\left(\text{QKTA}\right) = \text{Var}\left(\sum_{i=0}^{d} Q_{i,j}[t]\right) = \sum_{i=0}^{d} \text{Var}\left(Q_{i,j}[t]\right) = df_Q\left(1 - f_Q\right), \tag{16}$$

$$\text{E}\left(\text{QKCA}\right) = \text{E}\left(\sum_{j=0}^{N} Q_{i,j}[t]\right) = \sum_{j=0}^{N} \text{E}\left(Q_{i,j}[t]\right) = Nf_Q, \tag{17}$$

$$\text{Var}\left(\text{QKCA}\right) = \text{Var}\left(\sum_{j=0}^{N} Q_{i,j}[t]\right) = \sum_{j=0}^{N} \text{Var}\left(Q_{i,j}[t]\right) = Nf_Q\left(1 - f_Q\right), \tag{18}$$

where $d = D/h$ is the feature dimension of a head in the multi-head Q-K attention and $N$ is the token number.

**Mathematical Characteristics of SSA.** Similar to the above analysis process, assume that all elements $Q_{i,j}[t], K_{j,i}[t], V_{i,j}[t]$ in SSA are independent random variables and subject to Bernoulli distribution $B(f_Q), B(f_K), B(f_V)$, respectively. $f_Q, f_K$ and $f_V$ are the average firing rate of **Q**, **K** and **V**, respectively. We can calculate the expectation and variance of SSA as follows.

$$\text{E(SSA)} = \text{E}\left(\sum_{i=1}^{N}\sum_{j=1}^{d} Q_{i,j}[t]K_{j,i}[t]V_{i,j}[t]\right) = \sum_{i=1}^{N}\sum_{j=1}^{d} \text{E}\left(Q_{i,j}[t]K_{j,i}[t]V_{i,j}[t]\right) = Ndf_Qf_Kf_V, \tag{19}$$

$$\begin{aligned}
\text{Var(SSA)} &= \text{Var}\left(\sum_{i=1}^{N}\sum_{j=1}^{d} Q_{i,j}[t]K_{j,i}[t]V_{i,j}[t]\right) = \sum_{i=1}^{N}\sum_{j=1}^{d} \text{Var}\left(Q_{i,j}[t]K_{j,i}[t]V_{i,j}[t]\right) \\
&= Nd\left(f_Qf_Kf_V\left(1 - f_Q\right)\left(1 - f_K\right)\left(1 - f_V\right)\right. \\
&\quad + f_Qf_Kf_V^2\left(1 - f_Q\right)\left(1 - f_K\right) + f_Qf_K^2f_V\left(1 - f_Q\right)\left(1 - f_V\right) + f_Q^2f_Kf_V\left(1 - f_K\right)\left(1 - f_V\right) \\
&\quad \left. + f_Qf_K^2f_V^2\left(1 - f_Q\right) + f_Q^2f_Kf_V^2\left(1 - f_K\right) + f_Q^2f_K^2f_V\left(1 - f_V\right)\right),
\end{aligned} \tag{20}$$

Figure.4 visualize the variance and expectation of QKTA (Formula.15 and 16) and SSA (Formula.19 and 20). $N$ is set as 196 and $d$ is set as 64, respectively. We can find SSA has a much larger variance and expectation than QKTA on the whole. For example, the maximum theoretical variance of QKTA is 16, but the maximum theoretical variance of SSA is over 3000. This is the main reason that Q-K attention can discard scaling operations thus reducing power consumption, but SSA can not.

### 7.3 Futher Discussion on Q-K Attention

**The Complexity of Attention Mechanisms.** The computational complexity of SSA: $Q, K \in [0,1]^{N \times D}$. The attention map ($Q \times K^{\mathrm{T}} \in Z^{N \times N}$) is obtained by matrix multiplication of matrix $[0,1]^{N \times D}$ and matrix $[0,1]^{D \times N}$, which thus need $O(N^2 D)$ computation. The computational complexity of SDSA: $Q, K \in [0,1]^{N \times D}$. The attention map ($Q \otimes K \in [0,1]^{N \times D}$) is obtained by the Hadamard product of matrix $[0,1]^{N \times D}$ and matrix $[0,1]^{N \times D}$, which thus need $O(ND)$ computation. The computational complexity of Q-K Attention: Our attention vector ($A_t \in [0,1]^{N \times 1}$) is computed by $A_t = SN(\sum_{i=0}^{D} Q_{i,j})$, which depends on the row or column accumulation of the $Q$ matrix ($Q \in [0,1]^{N \times D}$), thus only needs $O(N)$ or $O(D)$ computation.

**PLIF for Scaling.** Q-K attention can discard scaling operations to reduce power consumption On these datasets used in this article's experiments because the variance of Q-K attention is much smaller than SSA (e.g. the max theoretical variance of Q-K token attention is only about 1 / 200 of SSA). For generality, we can also replace the LIF after attention calculation with PLIF (LIF with trainable parameters) allowing for adaptively controlling the fire rate of that spiking neuron, which can be seen as a learnable scaling. It can be expressed as $\mathbf{A}_t = \mathrm{PLIF}(\sum_{i=0}^{D} \mathbf{Q}_{i,j})$. The results show that this modification brings a 0.2% performance improvement on CIFAR 100 (Acc = 81.17%, the firing rate of $\mathbf{A}_t$ is 0.2952 in stage1 and 0.4008 in stage 2), while increasing the training time to 1.3 times.

### 7.4 Supplementary for Method 3.5

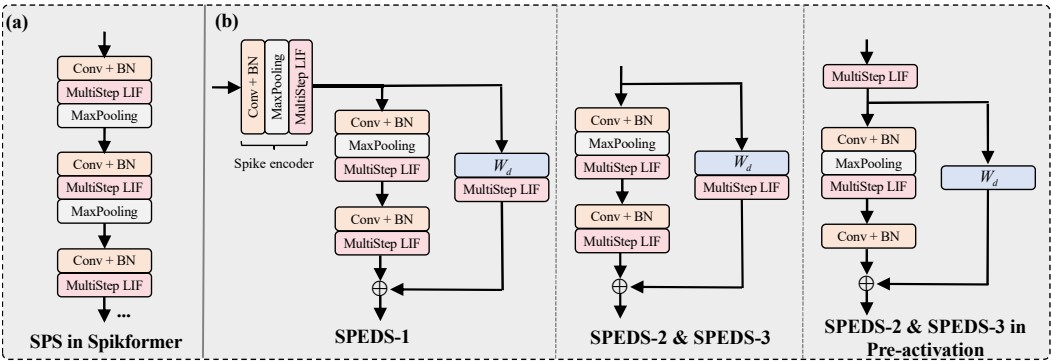

Figure 5: (a) Spiking Patch Splitting (SPS) module in Spikformer. (b) Spiking Patch Embedding with Deformed Shortcut (SPEDS) module in QKFormer.

### 7.5 Experimental Details

**Datasets.** We evaluate QKFormer on static image classification and neuromorphic classification. The former includes ImageNet-1K [39], CIFAR10/100 [40]. The latter contains CIFAR10-DVS [41] and DVS128 Gesture [42].

ImageNet-1K is the most typical static image dataset for classification. It contains 1.28 million images for training and 50k images for validation, with a total of 1,000 categories. CIFAR10/CIFAR100 provides 50, 000 train and 10, 000 test images with 32 × 32 resolution. The difference is that CIFAR10 contains 10 categories for classification. While CIFAR100 contains 100 categories, owning better distinguishing ability for the classification algorithm.

CIFAR10-DVS is an event-based neuromorphic dataset converted from the static image dataset by capturing shifting image samples through the Dynamic Vision Sensor (DVS) camera, which provides 9,000 training samples and 1,000 test samples. DVS128 Gesture is an event-based gesture

recognition dataset that contains 1342 samples of 11 hand gesture categories from 29 individuals under 3 illumination conditions, each gesture has an average duration of 6 seconds.

**Training Details.** In our experiments, we use 8 NVIDIA Tesla V100 SXM2 32GB GPUs when training models on ImageNet, while 1 GPU is used to train other datasets (CIFAR10, CIFAR100, DVS128 Gesture, CIFAR10-DVS). In direct training SNN models with surrogate function,

$$\sigma(x) = \frac{1}{1 + \exp(-\alpha x)}, \tag{21}$$

we select the Sigmoid function $\sigma(x)$ as the surrogate function with $\alpha = 4$ during the backpropagation of direct training in all experiments.

**Experimental Details on CIFAR and Neuromorphic Classification.** We evaluate our QKFormer on small-scale datasets, including CIFAR10, CIFAR100 [40] and temporal neuromorphic datasets (CIFAR10-DVS and DVS128 Gesture [42]). The detailed results on the four small-scale datasets are presented in Table 8.

Table 8: Comparision on CIFAR10, CIFAR100, DVS128, CIFAR10-DVS.

| Dataset | Methods | Architecture | Param (M) | Time Step | Top-1 Acc (%) |
|---|---|---|---|---|---|
| CIFAR10 | STBP-tdBN[43] | ResNet-19 | 12.63 | 4 | 92.92 |
| | TET[44] | ResNet-19 | 12.63 | 4 | 94.44 |
| | Spikformer[11] | Spikformer-4-384 | 9.32 | 4 | 95.51 |
| | Spikingformer[12] | Spikingformer-4-384 | 9.32 | 4 | 95.81 |
| | CML[14] | Spikformer-4-384 | 9.32 | 4 | 96.04 |
| | S-Transformer[13] | S-Transformer-2-512 | 10.28 | 4 | 95.60 |
| | **QKFormer** | HST-4-384 | 6.74 | 4 | **96.18** |
| CIFAR100 | STBP-tdBN[43] | ResNet-19 | 12.63 | 4 | 70.86 |
| | TET[44] | ResNet-19 | 12.63 | 4 | 74.47 |
| | Spikformer[11] | Spikformer-4-384 | 9.32 | 4 | 78.21 |
| | Spikingformer[12] | Spikingformer-4-384 | 9.32 | 4 | 78.21 |
| | CML[14] | Spikformer-4-384 | 9.32 | 4 | 80.02 |
| | S-Transformer[13] | S-Transformer-2-512 | 10.28 | 4 | 78.4 |
| | **QKFormer** | HST-4-384 | 6.74 | 4 | **81.15** |
| DVS128 | Spikformer[11] | Spikformer-2-256 | 2.57 | 10 , 16 | 96.9 , 98.3 |
| | Spikingformer[12] | Spikingformer-2-256 | 2.57 | 10 , 16 | 96.2 , 98.3 |
| | CML[14] | Spikformer-2-256 | 2.57 | 10 , 16 | 97.6 , 98.6 |
| | S-Transformer[13] | S-Transformer-2-256 | 2.57 | 16 | **99.3** |
| | STSA[15] | STSFormer-2-256 | 1.99 | 10 , 16 | 97.3 , 98.7 |
| | **QKFormer** | HST-2-256 | 1.50 | 10 , 16 | 98.3 , 98.6 |
| CIFAR10-DVS | Spikformer[11] | Spikformer-2-256 | 2.57 | 10 , 16 | 78.9 , 80.9 |
| | Spikingformer[12] | Spikingformer-2-256 | 2.57 | 10 , 16 | 79.9 , 81.3 |
| | CML[14] | Spikformer-2-256 | 2.57 | 10 , 16 | 79.2 , 80.9 |
| | S-Transformer[13] | S-Transformer-2-256 | 2.57 | 16 | 80.0 |
| | STSA[15] | STSFormer-2-256 | 1.99 | 10 , 16 | 78.96 , 79.93 |
| | **QKFormer** | HST-2-256 | 1.50 | 10 , 16 | 83.8 , **84.0** |

**Training and Testing Curve on ImageNet.** We show the training loss, test loss, top-1, and top-5 test accuracy of QKFormer (64.96M, 29.08M, 16.47M) on ImageNet-1K in Figure 6.

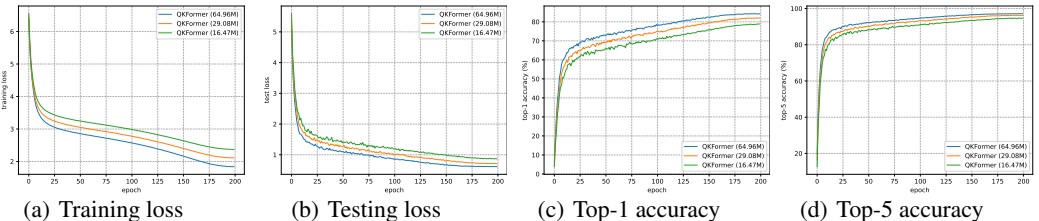

| (a) Training loss | (b) Testing loss | (c) Top-1 accuracy | (d) Top-5 accuracy |

Figure 6: Training loss, test loss, top-1 and top-5 test accuracy of QKFormer on ImageNet-1K. The input resolution of training and testing are $224 \times 224$.

## 7.6 Energy Consumption Calculation of QKFormer and ANNs

The homogeneity of convolution allows the following BN and linear scaling transformation to be equivalently fused into the convolutional layer with an added bias when deployment [45, 46, 47, 48]. Therefore, when calculating the theoretical energy consumption, the consumption of BN layers could be ignored. We calculate the number of Synaptic Operations (SOPs) of spike before calculating theoretical energy consumption for QKFormer.

$$\text{SOP}^l = fr \times T \times \text{FLOPs}^l \tag{22}$$

where $l$ is a block/layer in QKFormer, $fr$ is the firing rate of the block/layer and $T$ is the simulation time step of spiking neuron. $\text{FLOPs}^l$ refers to floating point operations of block/layer $l$, which is the number of multiply-and-accumulate (MAC) operations. And $\text{SOP}^l$ is the number of spike-based accumulate (AC) operations. Refer to previous works[49, 31, 50, 11, 12, 51, 52]. we assume that the MAC and AC operations are implemented on the 45nm hardware [50], where $E_{MAC} = 4.6pJ$ and $E_{AC} = 0.9pJ$. The theoretical energy consumption of QKFormer can be calculated as follows:

$$E_{\text{QKFormer}} = E_{AC} \times \left( \sum_{i=2}^{N} \text{SOP}_{\text{Conv}}^i + \sum_{j=1}^{M} \text{SOP}_{\text{QKTA}}^j + \sum_{k=1}^{Z} \text{SOP}_{\text{SSA}}^k \right) + E_{MAC} \times \left( \text{FLOP}_{\text{Conv}}^1 \right) \tag{23}$$

Eq.23 shows the energy consumption of QKFormer. $\text{FLOP}_{Conv}^1$ is the first layer encoding the non-spike input into spike-form. Then the SOPs of $N$ SNN Conv layers, $M$ QKTA layers, and $Z$ SSA layers are added together and multiplied by $E_{AC}$. For ANNs, the theoretical energy consumption can be calculated:

$$E_{\text{ANN}} = E_{MAC} \times \text{FLOPs} \tag{24}$$

## 7.7 Supplementary for Memory Consumption in Experiment 4.3

Table 9 shows the detailed values of Figure 4 in the main body of this paper (Experiment 4.3). We compare the memory consumption between QKTA (Formula.6) and SSA (Formula.4) under different token numbers, which is calculated on a GPU by forwarding the input tensor $(T, B, C, N)$. To facilitate the statistics of the impact of #tokens $N$ on memory consumption, the #channels $C$ are set to 256, and the time step $T$ and batch size $B$ are set to 1. The experiment result is shown in Figure 3(b). With the increment of #Tokens, SSA consumes much more GPU memory than QKTA, of which the complexity is linear to #Tokens. For example, SSA consumes about $10\times$ GPU memory than QKTA when $\sqrt{N} = 50$.

## 7.8 The Training and Inference Time Comparison.

We organized the experiment to test the training and inference time of QKFormer compared with Spikformer. We carried out this experiment on ImageNet with an input size of 224*224. This

Table 9: Detailed values of memory consumption of Figure 4.

| $\sqrt{N}$ | QKTA (M) | SSA (M) | SSA / QKTA |
|---|---|---|---|
| 10 | 0.10 | 0.14 | 1.37 |
| 20 | 0.40 | 1.00 | 2.53 |
| 30 | 0.89 | 3.97 | 4.46 |
| 40 | 1.58 | 12.19 | 7.71 |
| 50 | 2.47 | 26.44 | 10.70 |
| 60 | 3.56 | 53.52 | 15.04 |
| 70 | 4.84 | 97.64 | 20.17 |
| 80 | 6.32 | 162.50 | 25.70 |
| 90 | 8.18 | 258.19 | 31.55 |
| 100 | 10.35 | 391.77 | 37.85 |
| 110 | 12.14 | 570.69 | 47.01 |
| 120 | 14.23 | 806.06 | 56.66 |
| 130 | 16.70 | 1107.50 | 66.33 |
| 140 | 20.22 | 1485.14 | 73.43 |
| 150 | 22.26 | 1954.03 | 87.79 |
| 160 | 26.29 | 2525.00 | 96.03 |
| 170 | 28.55 | 3214.30 | 112.57 |
| 180 | 32.37 | 4036.88 | 124.71 |
| 190 | 36.41 | 5007.25 | 137.51 |
| 200 | 40.46 | 6143.06 | 151.84 |

experiment is carried out on a Ubuntu 18.04.6 LTS server with the Intel(R) Xeon(R) W-2125 CPU @ 4.00GHz, and the GeForce RTX 2080 (8G) GPU. "BS" means Batch Size. The experimental results are as follows:

Table 10: The training and inference time comparison between QKFormer and Spikformer.

| Model | Inference time (1 batch) | Training time (1 batch) |
|---|---|---|
| Spikformer(29.68M, T=4) , BS=6 | 1.63s | 2.65s |
| QKFormer(29.08M,T=4) , BS=6 | 1.82s | 3.62s |
| Spikformer(29.68M, T=4) , BS=1 | 1.46s | 2.08s |
| QKFormer(29.08M, T=4) , BS=1 | 1.33s | 2.72s |

In terms of inference time, QKFormer and Spikformer are very close. In terms of training time, QKFormer is about 1.35 times the training time of Spikformer in one batch, caused by hierarchical architecture. The training epochs of QKFormer on ImageNet are 200, while the training epochs of Spikformer are 300 [11], thus, the total training time cost of QKFormer on ImageNet is close to Spikformer's.

## 7.9 Discussion

**Prospect.** The human brain has powerful intelligence that runs with low power consumption, so how to develop novel artificial intelligence algorithms to achieve high performance with the low power consumption of the human brain level is one of the AI's ultimate goals. SNN is an attractive potential way to achieve it. QKFormer directly trained on ImageNet-1K has a groundbreaking leap forward with 10.84% accuracy improvement compared to the previous SNN model while maintaining the energy-efficient feature, which marks an important step towards this goal. Combined with pre-training in the future, the performance of QKFormer is expected to be further improved.

**Reproducibility.** The experimental results in this paper are reproducible. All experiments are implemented based on Pytorch [53], SpikingJelly [54] and Timm [55]. We explain the details of model training and configuration in the main text and Appendix. Our codes and models of QKFormer will be available on GitHub after review.

