# OpenReview forum: "QKFormer: Hierarchical Spiking Transformer using Q-K Attention"
_NeurIPS.cc/2024/Conference — NeurIPS 2024 spotlight_

### Official Review · Reviewer_7jT8 · 2024-07-11

**Soundness:** 3
**Presentation:** 3
**Contribution:** 3
**Rating:** 7
**Confidence:** 5

**Summary:**

The authors introduce a novel spiking transformer, QKFormer, which incorporates several innovative features: a spike-form Q-K attention mechanism with linear complexity and enhanced energy efficiency, a hierarchical structure that facilitates multi-scale spiking representation, and a spiking patch embedding with a deformed shortcut designed to optimize spiking information transmission and integration. Notably, this model achieves a remarkable top-1 accuracy of 85.65% on the ImageNet-1k dataset, marking the first instance where directly trained spiking neural networks have surpassed the 85% accuracy threshold on this benchmark.

**Strengths:**

1) The authors' introduction of a novel spike-form Q-K attention module with linear complexity represents a significant advancement in mitigating the quadratic computational complexity of spiking self-attention, a core component of traditional spiking transformers. This innovation demonstrates a substantial degree of originality and contributes to the optimization of spiking neural network architectures.
2) The authors' design of a powerful spiking patch embedding with deformed shortcut (SPEDS) is a notable achievement, as it effectively enhances spiking information transmission and integration. This enhancement contributes to the overall performance of the proposed QKFormer model and underscores the authors' creative approach to improving SNN performance.
3) The proposed QKFormer model, leveraging the innovative Q-K attention and SPEDS, represents a substantial leap forward in SNN performance. Its ability to outperform state-of-the-art SNNs on various static and neuromorphic datasets, particularly the ImageNet-1K dataset, where it achieves over 85% accuracy for the first time with directly trained SNNs, underscores its significance and potential impact on real-world applications of SNNs. This achievement not only demonstrates the model's quality but also highlights its potential to revolutionize the field of SNN research.

**Weaknesses:**

1) The authors' assertion that SPEDS can facilitate spiking information transmission and integration lacks sufficient elaboration on the underlying mechanisms.
2) Clarifying the rationale behind the assertion that the low variance of Q-K attention negates the necessity for scaling operations would significantly enhance the paper's rigor and credibility.
3) To improve the overall writing quality and grammar of the paper, several revisions are recommended. Firstly, in the caption of Figure 1, the phrase "The input is" should be revised to "The inputs are" to ensure consistency with the plural form of "inputs." Secondly, the expression in line 246 appears unusual and requires refinement to enhance clarity. Lastly, the abbreviation "LF" should be corrected to "IF" in line 316, assuming it is a typographical error and should stand for "if." Additionally, a comprehensive review and refinement of the entire paper's language, grammar, and structure would further elevate its academic style and readability.

**Questions:**

1) Could the summation operation in the QK attention mechanism potentially result in an excessively high firing rate within the attention vector?
2) Could you elaborate on the relationship between Figure 3a and the original input image?
3) As noted in the weaknesses section.

**Limitations:**

The authors briefly mention limitations in the Appendix, but discussing them in the main text would enhance clarity. Another limitation is that the model was only evaluated on image classification, limiting its generalizability.

---

> ### Author Rebuttal · Authors · 2024-08-06
>
> Thanks for your insightful comments. We have carefully studied your comments and have made every effort to address your concerns. We will include the relevant analysis in the revised manuscript accordingly.
>
> ### **Weaknesses**
> > ***Weaknesses 1**: The authors' assertion that SPEDS can facilitate spiking information transmission and integration lacks sufficient elaboration on the underlying mechanisms.*
>
> **WA1:** Residual shortcuts in SNNs [1] can implement identity mapping, which reduces information loss (facilitates information transmission and integration) in spike communication, thus ensuring the network can be well-behaved in a depth-insensitive way. Previous spiking transformers [2, 3] use the residual shortcuts to achieve identity mapping, mainly focusing on the spiking Attention block and spiking MLP block, and lacking identity mapping in patch embedding across the downsampling block. **In SEPDS, we perform a lightweight linear projection $\mathbf{W}_d$ in the shortcut connections to match the channel (and token) numbers, thus realizing the identity mapping cross downsampling blocks in spiking patch embedding.**
>
> > ***Weaknesses 2**: Clarifying the rationale behind the assertion that the low variance of Q-K attention negates the necessity for scaling operations would significantly enhance the paper's rigor and credibility.*
>
> **WA2:** Self-attention (sec. 3.2.1) [4] shows the variance magnitude of ${Q} {K}^{\mathrm{T}}$ grows with the embedding dimension $d$,  which can result in gradient vanishing issues after the softmax operation. In other words, if the input magnitude is very large, the gradient of softmax will tend to 0, causing the gradient to disappear. **The larger the variance, the more likely the dot product result is larger in magnitude.** Therefore, the product of matrices ${Q}$ and ${K}$ in self-attention is scaled by a factor $\frac{1}{\sqrt{d}}$ to normalize the product to variance 1.  In contrast, SSA-based [3] SNNs are prone to suffer from performance degradation, and even cannot converge without scaling, because the variance of ${Q} {K}^{\mathrm{T}} {V}$ output is too large for LIF neuron with surrogate gradient method. However, Q-K attention can discard scaling operations and reduce power consumption because the variance of Q-K attention is much smaller than SSA (e.g. the max theoretical variance of Q-K token attention is only about 1 / 200 of SSA, see Sec. 3.3 and 4.3 and Appendix 6.2 in the manuscript).
>
> > ***Weaknesses 3**: To improve the overall writing quality and grammar of the paper, several revisions are recommended. ... a comprehensive review and refinement of the entire paper's language, grammar, and structure would further elevate its academic style and readability.*
>
> **WA3:** Sorry for this and we sincerely thank you for your detailed suggestions, which are very helpful for us to improve the quality of our manuscript. We have made the following revisions:
> 1) "The input are" in Figure 1 has been changed to "The inputs are".
> 2) line 246 "Results on CIFAR and Temporal Neuromorphic Classification" has been changed to "Results on CIFAR and Neuromorphic Datasets".
> 3) line 316 "with LF and PLIF" has been changed to "with Integrate-and-Fire (IF)  and Parametric Leaky Integrated-and-Fire (PLIF)".
>
> In addition, we have conducted a comprehensive grammar and structure review of the entire manuscript.
>
> ### **Question**
>
> > ***Question 1**: Could the summation operation in the QK attention mechanism potentially result in an excessively high firing rate within the attention vector?*
>
> **QA1:** Actually, the summation operation in the Q-K attention will lead to $Q$ becoming very sparse compared to $K$ when the network converges. **As shown in Table R3 in "Global Rebuttal"**, the $Q$ in stage 1 has a fire rate of 0.0432, while $K$ has 0.1784. After the accumulation operation along $D/h$ of the multi-head QKTA version, **the LIF ($A\_t$) has a normal fire rate of 0.3477**. In addition, we could replace that LIF with PLIF to adaptively control the fire rate of that spiking neuron. The result shows that this way only brings a 0.2% performance improvement on CIFAR 100 (Acc = 81.17%, the firing rate of PLIF ($A_{t}$) is 0.2952 in stage1 and 0.4008 in stage2), but the training time will become about 1.3 times compared with the previous.
>
> > ***Question 2**: Could you elaborate on the relationship between Figure 3a and the original input image?*
>
> **QA2:** Figure 3a is mainly to visualize the firing state of Q-K attention in the network. We randomly select an image (224\*224) from ImageNet test set for inference and visualized ${{A}}_t$, ${K}$ and ${X}^{\prime}$ in QKTA. Input image: 224\*224 —> Stage1 feature: 56\*56 (flatten to 3136 tokens) —> Stage2 feature: 28\*28 (flatten to 784 tokens). Figure 3a chooses the continuous token segment with a length of 100 ([1:100] from [1:3136] in stage 1 and [1:784] in stage 2) to visualize the firing state in QKTA.
>
> ### Limitations
>
> > ***Limitations 1**: limitation discussion*
>
> **LA1:** According to your thoughtful comments, the limitation discussion has been changed to "The limitation discussion has been changed to “Currently, our model is limited to image/DVS classification tasks. We will extend this work to more tasks, such as segmentation, detection, and language tasks, to test the generalizability in the further. In addition, we will explore efficient and high-performance network architectures with fewer time steps based on Q-K attention and other efficient modules, to further reduce the training consumption.” We will add this discussion to the main text in the revised manuscript." We will add this discussion to the main text in the revised manuscript.
>
> [1] Deep Residual Learning in Spiking Neural Networks. In NeurIPS, 2021.
>
> [2] Spikformer: When spiking neural network meets transformer. In ICLR, 2023.
>
> [3] Spike-driven transformer. In NeurIPS, 2023.
>
> [4] Attention is all you need. In NeurIPS, 2017.

---

> > ### Comment · Reviewer_7jT8 · 2024-08-12
> >
> > I have reviewed the rebuttal and have decided to maintain my positive rating.

---

### Official Review · Reviewer_Etfg · 2024-07-11

**Soundness:** 3
**Presentation:** 3
**Contribution:** 2
**Rating:** 5
**Confidence:** 5

**Summary:**

This model introduces a spike-form Q-K attention mechanism that efficiently models the importance of token or channel dimensions using binary values, significantly reducing computational complexity .The model is evaluated on the ImageNet-1K dataset, achieving an impressive top-1 accuracy of 85.65%, surpassing existing benchmarks for spiking transformers.

**Strengths:**

1.	The proposed spike-form Q-K attention mechanism provides an efficient way to handle token or channel dimensions with binary spikes.
2.	The model achieves over 85% accuracy on ImageNet-1K, which is a very impressive result.

**Weaknesses:**

1、The Q-K attention module discussed in the paper has already been introduced in Spike-Driven Transformer V2[1] (SDSA-2 in Figure 3). Additionally, the overall architecture of the QKFormer model closely resembles that in [2].
2、 In Section 4.1, the authors incorrectly compare the full-precision results of their work with Binary Neural Networks. Upon reviewing the original texts, it was found that PokeBNN (20.7MB) [2] and MeliusNet59 (17.4MB) [3] report model sizes, not parameter counts (20.7M, 17.4M). Comparisons should use consistent metrics.
3、The baseline for the Q-K attention ablation study (QKCA + SSA) is not reasonable. There is no information on the model with only SSA or without the attention module.
4、The effectiveness of the SPEDS module with only Q-K attention and without SPEDS remains unclear. It appears that the architecture of Spikformer differs from QKFormer, with QKFormer’s architecture being more similar to that in Spike-Driven Transformer V2

[1] Man Yao, JiaKui Hu, Tianxiang Hu, Yifan Xu, Zhaokun Zhou, Yonghong Tian, XU Bo, and Guoqi Li. Spike-driven transformer v2: Meta spiking neural network architecture inspiring the design of next-generation neuromorphic chips. In The Twelfth International Conference on Learning Representations, 2023.
[2] Zhang Y, Zhang Z, Lew L. Pokebnn: A binary pursuit of lightweight accuracy[C]//Proceedings of the IEEE/CVF Conference on Computer Vision and Pattern Recognition. 2022: 12475-12485.
[3] Bethge J, Bartz C, Yang H, et al. Meliusnet: An improved network architecture for binary neural networks[C]//Proceedings of the IEEE/CVF Winter Conference on Applications of Computer Vision. 2021: 1439-1448.

**Questions:**

1.	The paper suggests that Q-K attention proposed here is simpler compared to SDSA-3 in Spike-Driven Transformer V2, and ablation experiments show performance degradation with Q-K Attention. The QKFormer architecture closely resembles [1] yet achieves a 5.65% improvement on ImageNet. Could this improvement be attributed to superior data augmentation or training strategies rather than the effectiveness of the attention methods and model architecture?
2.	Since the accuracy on ImageNet-1K significantly exceeds those of existing comparable models, it is challenging to evaluate the results without access to the corresponding code. Could you please make the code open source?

---

> ### Author Rebuttal · Authors · 2024-08-06
>
> Thanks for your insightful feedback and your time in reading our manuscript. We hope that the responses below could address your concerns. We will include the relevant analysis in the revised manuscript accordingly.
>
> ### **Weaknesses**
> > ***Weaknesses 1**: The Q-K attention module in the paper has already been introduced in Spike-Driven Transformer V2[1] (SDSA-2 in Figure 3). Additionally, the overall architecture of the QKFormer closely resembles that in [2]. **Question 1**: The paper suggests that Q-K attention here is simpler compared to SDSA-3 in Spike-Driven Transformer V2, and ablation experiments show performance degradation with Q-K Attention. The QKFormer architecture closely resembles [1] yet achieves a 5.65% improvement on ImageNet. Could this improvement be attributed to superior data augmentation or training strategies rather than the effectiveness of the attention methods and model architecture?*
>
> **WA1 & WQ1:**
> Thanks for your careful reading and comments! There are several major differences in our attention compared to SDSA-2[1], and we have other designs like SPEDS：
> 1) **In terms of attention**, we have proposed a **mixed spiking attention framework in hierarchical architecture for integration**, while Spike-Driven Transformer V2 uses single-dimension attention. **QKFormer (QKTA + SSA)** serves as the primary model tested in the main experiments on ImageNet-1K, CIFAR, and neuromorphic datasets. The mixed spiking attention framework (including QKTA + SSA and other mixed attention variants, as detailed in "2. Ablation Study of Q-K Attention" in the document "Global Rebuttal") could efficiently leverage the importance across multiple dimensions and improve the performance compared with SDSA-2, but with higher computation efficiency compared with only SSA. Therefore, our mixed attention strikes a balance between high performance and computing requirements in QKFormer architecture, enabling model size enlargements and performance improvements on the ImageNet-1K dataset.
>
> 2) **In terms of the synaptic computing layer of attention**, Q-K Attention uses a straightforward, deployment-friendly linear layer.  In contrast, SDSA-2 employs re-parameterization convolution sequences (conv->bn->conv->conv->bn layers), which necessitate tedious post-processing conversion to achieve a truly spike-driven SNN model.
>
> 3) **Other differences in architecture:** a new patch embedding module (SPEDS in Sec3.5 and 4.4 in the manuscript), hierarchical stages, etc.
>
> **From the results of the ablation study (Table R1 and Table R2) and their following discussions, our performance improvement mainly comes from the mixed attention, SPEDS, and overall hierarchical architecture.**
>
> **Training details:** For training strategy, we adopt a direct training method following Spikformer [2]. For data augmentation, we follow DeiT [3], using RandAugment [4], random erasing [5], and stochastic depth [6].
>
> In addition, the earliest version of Spike-Driven Transformer V2 was released on Arxiv on Feb 15, 2024, while ours was released on March 25, 2024. This indicates that both studies were conducted concurrently.
>
>
> **Finally, we would appreciate it if you could further clarify the similarities between QKFormer and PokeBNN [7], as mentioned in Weakness 1. This will help us address your concerns more effectively and provide a more thorough response. Thank you!**
>
>
> > ***Weaknesses 2**: In Sec 4.1, the authors incorrectly compare the full-precision results of their work with BNNs. It was found that PokeBNN (20.7MB) [2] and MeliusNet59 (17.4MB) [3] report model sizes, not parameter counts (20.7M, 17.4M). Comparisons should use consistent metrics.*
>
> **WA2:** We apologize for the oversight and will update the manuscript later.  We found that both PokeBNN [7] and MeliusNet59 [8] haven't reported the model parameters, but instead reported the number of computation operations. The comparison is as follows, and the operation numbers are comparable:
>
> PokeBNN 2.0x (77.2\%, BNN SOTA)[7]:14.412G binary operations, 0.0145G Int4_MACs, and 0.0107G Int8_MACs.
>
> MeliusNet-59 (71.0\%, BNN)[8]:18.3G binary operations and 0.245G MACs
>
> QKFormer-10-384(78.80\%, SNN): 15.12G ACs and 0.26G MACs.
>
>
> > ***Weaknesses 3**: The baseline for the Q-K attention ablation study (QKCA + SSA) is not reasonable. There is no information on the model with only SSA or without the attention module.**Weaknesses 4**: The effectiveness of the SPEDS module with only Q-K attention and without SPEDS remains unclear. It appears that the architecture of Spikformer differs from QKFormer, with QKFormer’s architecture being more similar to that in Spike-Driven Transformer V2.*
>
>
> **WA3 & WA4:**  Thanks for your constructive advice, which has greatly helped improve the quality of our manuscript. **Please refer to "Global Rebuttal" for responses to Weaknesses 3 and Weaknesses 4**.  According to your thoughtful comments, we add another two comparison models: QKFormer(w/o SPEDS) and QKFormer(SSA). In addition, we add the ablation study on
>  CIFAR10-DVS. The results indicate the effectiveness of mixed attention and SPEDS on both static and DVS datasets.
>
> ### **Question**
>
> > ***Question 2**: code open source?*
>
> **WQ2:** We will open the code as soon as the manuscript is accepted.
>
>
> [1] Spike-driven transformer v2: Meta spiking neural network architecture inspiring the design of next-generation neuromorphic chips. In ICLR 2024.
>
> [2] Spikformer: When spiking neural network meets transformer. In ICLR, 2023.
>
> [3] Training data-efficient image transformers & distillation through attention. In ICML, 2021.
>
> [4] Randaugment: Practical automated data augmentation with a reduced search space. In CVPR workshop, 2020.
>
> [5] Random erasing data augmentation. In AAAI, 2020.
>
> [6] Deep networks with stochastic depth. In ECCV, 2016.
>
> [7] Pokebnn: A binary pursuit of lightweight accuracy, In CVPR, 2022.
>
> [8] Meliusnet: An improved network architecture for binary neural networks. In WACV，2021.

---

> > ### Comment · Reviewer_Etfg · 2024-08-12
> >
> > I would like to thank the authors for addressing some of my concerns. I want to keep my score.

---

### Official Review · Reviewer_ukE8 · 2024-07-16

**Soundness:** 3
**Presentation:** 3
**Contribution:** 3
**Rating:** 6
**Confidence:** 4

**Summary:**

The author has thoughtfully considered the attention structure of the existing Spiking Transformer (SSA) as well as issues present in other parts. Spiking neural networks are characterized by their high efficiency and energy-saving features. For this purpose, the author has proposed a more efficient attention mechanism. To better enhance performance, the author has also made modifications to the embedding part. From the experimental results, the new model achieved better performance on several typical datasets with lower energy consumption and fewer parameters. Additionally, the ablation experiments designed by the author demonstrated the rationality of the attention mechanism and embedding modifications. Overall, the method proposed in the article is highly feasible and logically coherent.

**Strengths:**

1. The article analyzes the issue of computational redundancy in existing Spiking Transformer attention mechanisms.
2. The Q-K Attention offers an almost perfect solution, being simple in design (not a bad thing), yet reducing computational complexity while enhancing performance.
3. A better embedding method has been designed.
4. Ablation experiments have demonstrated the rationality of these approaches.

**Weaknesses:**

1. Lack of an overview of pipeline. Some charts, such as Figure 3, seem to be of little significance. In contrast, SPEDS might require a more intuitive representation, given that you have made significant modifications to the previous embedding.
2. Could the ablation study include additional experiments? We have seen experiments on SPEDS and Q-K Attention using CIFAR-100. Could we conduct a similar set of experiments on neuromorphic datasets? Given the unique nature of events, we hope to see SPEDS achieve performance improvements on neuromorphic datasets as well.

**Questions:**

I ampleased to see that such a simplified attention mechanism can improve performance while saving energy. But I still have the following questions:
1. The capability of SPEDS on event datasets needs to be demonstrated.
2. As shown Eq.6, row and column-wise summation is performed when computing Q-K attention. Attention mechanisms are designed to integrate global attention. Would Q-K attention cause the loss of features that do not align with the row and column directions?
3. After summation in Eq.6,, Q undergoes neuron processing, is this to ensure the model is spike-driven? Representations in SNNs are inherently sparse; would this lead to feature loss?

**Limitations:**

See weaknesses and Questions

---

> ### Author Rebuttal · Authors · 2024-08-06
>
> Thanks for your insightful comments and your time in reading our paper.  We have carefully studied your comments and have made every effort to address your concerns. We will include the relevant analysis in the revised manuscript accordingly.
>
> ### **Weaknesses**
> > ***Weaknesses 1**: Lack of an overview of pipeline. Some charts, such as Figure 3, seem to be of little significance. In contrast, SPEDS might require a more intuitive representation, given that you have made significant modifications to the previous embedding.*
>
> **WA1:** Thanks for your constructive advice! Please refer to the PDF in "Global Rebuttal", where we have added a more detailed pipeline including the overall architecture and all modules of QKFormer (Figure R1 in the uploaded pdf file). We will continue to improve the figure and include it in the revised manuscript accordingly.
>
>
> > ***Weaknesses 2**: Could the ablation study include additional experiments? We have seen experiments on SPEDS and Q-K Attention using CIFAR-100. Could we conduct a similar set of experiments on neuromorphic datasets? Given the unique nature of events, we hope to see SPEDS achieve performance improvements on neuromorphic datasets as well.*
> > ***Questions 1**: The capability of SPEDS on event datasets needs to be demonstrated.*
>
> **WA2 & WQ1:** Please refer to "Global Rebuttal". We add ablation experiments on the neuromorphic dataset: CIFAR10-DVS. In addition, we add another two comparison models:  QKFormer(w/o SPEDS) and QKFormer(SSA). The experimental results are shown in **Table R1** and **Table R2**, with the corresponding discussion provided following these tables.
>
> **SPEDS module on neuromorphic dataset**: The results show that SPEDS leads to general performance improvements on the CIFAR10-DVS dataset. a. The SPEDS module is essential to QKFormer on both static and neuromorphic datasets. b. Adding SPEDS to Spikformer brings great gains in CIFAR100 (+2.05%) and CIFAR10-DVS (+1.30%).
>
>
>
>
> ### **Questions**
> > ***Questions 2**: As shown Eq.6, row and column-wise summation is performed when computing Q-K attention. Attention mechanisms are designed to integrate global attention. Would Q-K attention cause the loss of features that do not align with the row and column directions?*
> > ***Questions 3**: After summation in Eq.6, Q undergoes neuron processing, is this to ensure the model is spike-driven? Representations in SNNs are inherently sparse; would this lead to feature loss?*
>
> **QA2 & QA3:** **The reviewer correctly notes that neuron processing after summation ensures the model is spike-driven.** By adding the spiking neuron in Eq. 6, the output in Figure 1 will be a spike-based representation rather than an integer matrix. This inclusion prevents non-spike computation between integer outputs and float weights in the post-linear layer.
>
>
> **Q-K attention models the importance of token or channel dimensions.** In other words, it identifies which token or channel is more important effectively. In contrast, SSA models the importance relationship between each two tokens. As you mentioned, a single type of Q-K attention may cause the loss of features that do not align with the row and column directions. So we proposed **the mixed spiking attention solution of QKFormer**, such as QKFormer(QKTA + QKCA), QKFormer(QKTA + SSA), and QKFormer(QKCA + SSA) with high performance. Please refer to Table R.2 and the corresponding discussion in "Global Rebuttal" (2. Ablation Study of Q-K Attention) for details.

---

> > ### Comment · Reviewer_ukE8 · 2024-08-11
> >
> > Thanks for your rebuttal. I keep my score.

---

### Official Review · Reviewer_m2NM · 2024-07-17

**Soundness:** 4
**Presentation:** 3
**Contribution:** 4
**Rating:** 8
**Confidence:** 5

**Summary:**

The authors proposed QKFormer, which pushes SNN to 85% accuracy in ImageNet, becoming a new sota and contributing to the SNN community.

**Strengths:**

1. The authors propose a model accuracy of Sota, which is 10% higher than the previous spikformer, and the number of parameters is lower than spikformer, which is undoubtedly the leader of the SNN-Transformer series.
2. The description is clear and the comparison is complete. The working mechanism of Q-K Attention is clearly expressed. The visualization results are rich. Comparison experiments are complete, comparing the time complexity and space complexity of different attentions in SNN domain in recent years, comparing with the Transformer architecture in ANN domain, covering the commonly used static datasets CIFAR10, CIFAR100, ImageNet, and DVS datasets DVS128, CIFAR10-DVS.

**Weaknesses:**

1. The current accuracy is high, but it still requires a large time step and a large amount of computation.

**Questions:**

1. In Table2 the authors list a comparison of multiple metrics for different models, but there is a large time step for SNN, how does QKFormer's time during training or inference compare to other models?
2. The authors could add an explanation of Table1 to the supplemental material to visually analyze the complexity of other attention mechanisms.
3. Q-K token attention Sum each column and then use LIF on the column vectors, as shown in Fig. 1(a), 4 to 1,2 to 0. How is the threshold set?

**Limitations:**

See Questions and Weaknesses.

---

> ### Author Rebuttal · Authors · 2024-08-05
>
> Thank you for your insightful feedback. We have carefully studied your comments and have made every effort to address your concerns. We will include the relevant analysis in the revised manuscript accordingly.
>
>
>
> ### **Weaknesses**
> > ***Weaknesses 1**: The current accuracy is high, but it still requires a large time step and a large amount of computation.*
>
> **WA1**: QKFormer achieves SOTA performance by keeping the same time step settings as the previous direct trained spiking transformers [1,2,3] (such as $T=4$ on ImageNet and CIFAR dataset). Further reducing the time step is worth exploring in SNN and your feedback will inspire our future work.
>
>
> ### **Questions**
> > ***Questions 1**: In Table 2 the authors list a comparison of multiple metrics for different models, but there is a large time step for SNN, how does QKFormer's time during training or inference compare to other models?*
>
> **QA1**: Thank you for your insightful question! To address your concern, we have tested the training and inference time of QKFormer and Spikformer for comparison [1]. We carry out this experiment on ImageNet with an input size of 224*224, on an Ubuntu 18.04.6 LTS server with an Intel(R) Xeon(R) W-2125 CPU @ 4.00GHz, and a GeForce RTX 2080 (8G) GPU. "BS" means Batch Size. The experimental results are as follows:
>
>
> **Table R4: The training and inference time comparison**
>
> | Model | Inference time (1 batch) | Training time (1 batch)|
> | -------- | -------- | -------- |
> | Spikformer(29.68M, T=4) , BS=6  | 1.63s     | 2.65s    |
> | QKFormer(29.08M,T=4) , BS=6  | 1.82s      | 3.62s    |
> | Spikformer(29.68M, T=4) , BS=1 | 1.46s      | 2.08s   |
> | QKFormer(29.08M, T=4) , BS=1 | 1.33s      | 2.72s    |
>
> **In terms of inference time**, QKFormer and Spikformer perform comparably. **In terms of training time**, QKFormer requires approximately 1.35 times the duration per batch compared to Spikformer, due to its hierarchical architecture. **In terms of the training epochs**, QKFormer is trained on ImageNet for 200 epochs, while Spikformer requires 300 epochs [1]. Consequently, **the total training time cost of QKFormer on ImageNet is close to Spikformer's**.
>
>
>
> > ***Questions 2**: The authors could add an explanation of Table1 to the supplemental material to visually analyze the complexity of other attention mechanisms.*
>
> **QA2**: Thank you for your suggestion! We explain Table1 in detail as follows and will add it to the revised manuscript accordingly.
>
> The computational complexity of **SSA**: $Q, K \in [0, 1]^{N \times D}$. The attention map ($Q \times K^{\mathrm{T}} \in Z^{N \times N}$) is obtained by matrix multiplication of matrix $[0, 1]^{N \times D}$ and matrix $[0, 1]^{D \times N}$, which requires $O(N^2 D)$ computation.
>
> The computational complexity of **SDSA**: $Q, K \in [0, 1]^{N \times D}$. The attention map ($Q \otimes K \in [0, 1]^{N \times D}$ ) is obtained by the Hadamard product of matrix $[0, 1]^{N \times D}$ and matrix $[0, 1]^{N \times D}$, which requires $O(ND)$ computation.
>
> The computational complexity of **Q-K Attention**: Our attention vector (${A}_t \in [0, 1]^{N \times 1}$) is computed by
>
> ${A}\_t = SN(\sum_{i=0}^D {Q}_{i, j})$,
>
> which depends on the row or column accumulation of the $Q$ matrix ($Q \in [0, 1]^{N \times D}$), thus only requires $O(N)$ or $O(D)$ computation.
>
>
> > ***Questions 3**: Q-K token attention Sum each column and then use LIF on the column vectors, as shown in Fig. 1(a), 4 to 1,2 to 0. How is the threshold set?*
>
> **QA3**: The LIF neuron is implemented by Spikingjelly [4]. The time constant is set as 2, the threshold is 0.5 by default. You may worry that the accumulation operation along $D/h$ will lead to the following LIF over-fires.
>
> i) Actually, **the hyperparameter settings of LIF in our work will lead to $Q$ becoming very sparse compared to $K$ when the network converges**. **see Table R3**, **the $Q$ in stage 1 has a fire rate of 0.0432, while $K$ has 0.1784**. After the accumulation operation along $D/h$ of the multi-head QKTA version, **the LIF ($A\_t$) has a typical fire rate of 0.3477**.
>
> ii) In addition, we have carried out an experiment that replaced that LIF with PLIF (LIF with trainable parameters) to tune the fire rate of that spiking neuron adaptively. The result shows that this way only brings a 0.2% performance improvement on CIFAR 100 (Acc = 81.17%, the firing rate of PLIF($A\_t$) is 0.2952 in stage1 and 0.4008 in stage2), but the training time will become about 1.3 times compared with the LIF version.
>
> We will include the details in the revised manuscript accordingly.
>
>
> [1] Spikformer: When spiking neural network meets transformer. In ICLR, 2023.
>
> [2] Spikingformer: Spike-driven residual learning for transformer-based spiking neural network. In Arxiv, 2023.
>
> [3] Spike-driven transformer. In NeurIPS, 2023.
>
> [4] SpikingJelly: An open-source machine learning infrastructure platform for spike-based intelligence. In Science Advance, 2023.

---

> > ### Comment · Reviewer_m2NM · 2024-08-14
> >
> > The author answered my confusion and therefore raised the score.

---

### Author Rebuttal · Authors · 2024-08-06

Dear ACs and Reviewers,

We would like to first express our gratitude to all the reviewers for their valuable comments. We are encouraged that reviewers have commended the performance of QKFormer architecture and its components including Q-K Attention (Q-K Token Attention and Q-K Channel Attention) and Spiking Patch Embedding with Deformed Shortcut (SPEDS) modules. Meanwhile, most reviewers are concerned about the ablation study of Q-K attention and SPEDS module, and whether the accumulation mechanism of Q-K attention leads to over-fire of spiking neurons. Our responses to these questions are as follows. We will include the relevant analysis in the revised manuscript accordingly.

### **1. Ablation Study for SPEDS Module：**

Based on the ablation results presented in Table 4 of the manuscript, we have added:

a. another model, QKFormer(w/o SPEDS); b. the ablation study on the neuromorphic dataset, CIFAR10-DVS. The experimental results are as follows:

**Table R1: Ablation studies for SPEDS.**
| Model | CIFAR100 Acc  |CIFAR10-DVS Acc|
| -------- | -------- | -------- |
| QKFormer(QKTA+SSA)| 81.15%  | 84.00%|
| QKFormer(QKTA+SSA, w/o SPEDS) | 80.08% |  83.40%|
| Spikformer(SSA) | 78.21% |  80.90%|
| Spikformer(SSA) + SPEDS | 80.26% |  82.20%|

The results show that the SPEDS module is essential to QKFormer on both static and neuromorphic datasets. In addition, the addition of SPEDS to Spikformer leads to great gains in CIFAR100 (+2.05%) and CIFAR10-DVS (+1.30%), which further verified the effectiveness of SPEDS.

### **2. Ablation Study of Q-K Attention：**

Based on the ablation results presented in Table 4 of the manuscript, we have added:

a. another model, QKFormer(SSA); b. the ablation study on the neuromorphic dataset, CIFAR10-DVS. The experimental results are as follows:


**Table R2: Ablation studies for Q-K Attention.**
| Model | CIFAR100 (Acc, Param) |CIFAR10-DVS (Acc, Param)|
| -------- | -------- | -------- |
| QKFormer(QKTA+SSA, Baseline)| 81.15%, 6.70M  | 84.0%, 1.50M|
| QKFormer (QKCA + SSA) | 81.07%, 6.70M       | 84.30%, 1.50M|
| QKFormer (QKTA + QKCA) | 81.04%, 6.44M      | 83.10%, 1.44M|
| QKFormer (SSA ) | 81.23%, 6.79M             |  84.10%, 1.52M|
| QKFormer (QKCA) | 81.00%, 6.44M             |  81.50%, 1.44M|
| QKFormer (QKTA) | 79.09%, 6.44M             |  80.70%, 1.44M|

**1) QKFormer(SSA):**

The results in the table above clearly demonstrate **the superiority of our QKFormer architecture: fewer parameters, much stronger performance**. For instance, CIFAR100: QKFormer(SSA, 6.79M, 81.23%) VS. Spikformer(SSA, 9.32M, 78.21%) and CIFAR10-DVS: QKFormer(SSA, 1.52M, 84.10%) VS. Spikformer(SSA, 2.57M, 80.90%). **Moreover, applying the hierarchical structure with SSA, QKFormer(SSA), on large datasets (such as ImageNet-1k with 224 * 224 input size) poses challenges due to its high requirements of computational and memory resources.** For example, due to memory explosion, QKFormer(SSA) cannot be trained on ImageNet even with a batch size of 1 on an NVIDIA Tesla V100 GPU (32G). This limitation of SSA is also a key reason for us to propose the Q-K attention module with reduced computational complexity.


**2) Mixed spiking attention integration：**

**Through our exploration, the mixed spiking attention of QKFormer is the optimal solution when considering both computational efficiency and performance.** Using a single type of Q-K attention leads to reduced performance. Mixed spiking attention solutions, such as QKFormer(QKTA + QKCA), QKFormer(QKTA + SSA), and QKFormer(QKCA + SSA) can achieve comparable performance to QKFormer(SSA) while requiring fewer parameters and much fewer memory resources (Figure 3b and Table 7 in the manuscript). Consequently, the mixed spiking attention solutions are well-suited for larger architectures and more challenging scenarios."


### **3. Spike Firing Rates in Q-K Attention (QKTA)：**
We have calculated the spike firing rates for the QKFormer blocks of the trained QKFormer (64.9M) on ImageNet-1k and included them in Appendix A.5 as Table 6. We apologize for the lack of sufficient discussion. The average spike firing rates of the QKTA across neurons and time steps in Stage1 and Stage2 are also as follows:

**Table R3: Spike firing rates in Q-K Attention (QKTA) on ImageNet. $A_{t}$ denotes the fire rate of the LIF neuron after accumulation operation.**
| QKTA | Stage1(firing rate) |Stage2(firing rate)|
| -------- | -------- | -------- |
| $Q$| 0.0432  | 0.0231|
| $K$ | 0.1784       | 0.0847|
| $A_{t}$ | 0.3477      | 0.2655|
| ${X}^{\prime}$ | 0.0832       | 0.0350|
| ${X}^{\prime\prime}$ | 0.1478       | 0.0577|


In fact, the summation operation in the Q-K attention causes **$Q$ to become significantly sparser compared to $K$ when the network converges**. Specifically, $Q$ in stage 1 has a fire rate of 0.0432, while $K$ has 0.1784. **After the accumulation operation along $D/h$ of the multi-head QKTA version, the LIF neuron ($A_{t}$) exhibits a typical averaged fire rate of 0.3477.** In addition, replacing the LIF with PLIF (LIF with trainable parameters) allows for adaptively controlling the fire rate of that spiking neuron. The results show that this modification only brings a 0.2% performance improvement on CIFAR 100 (Acc = 81.17%, the firing rate of PLIF($A_{t}$) is 0.2952 in stage1 and 0.4008 in stage2), while increasing the training time to 1.3 times.


Best regards

Authors

---

### Decision · Program_Chairs · 2024-09-25

**Decision:**

Accept (spotlight)

**Comment:**

Spiking neural networks that more closely resemble real neurons have long been of interest as a more biologically grounded alternative to deep networks, but their performance have long lagged behind. In this paper, a spiking neural network transformer model is presented that achieves a large improvement above the state of the art in spiking neural networks. Specific design decisions also make this model very efficient along with being very accurate (such as a new efficient spike-based attention module that allows the construction of larger models, a multi-scale spiking architecture that models a hierarchy of representations, a new embedding approach). The paper is well written and describes the approach clearly. The authors perform a comprehensive analysis of the time and space complexity of recent spiking neural network architecture on the main datasets typically used in such approaches, effectively comparing their approach in a sound way to existing work. The author also perform informative ablation analysis. The analyses in the paper were appreciated by the reviewers, who also requested additional clarifications and analyses. All these modifications should be added to the paper for the camera ready version.